

# Boundary layer moisture variability at the ARM Eastern North Atlantic Observatory

Maria P. Cadeddu[1], Virendra P. Ghate[1], David D. Turner[2], Thomas E. Surleta[3]

[1]Argonne National Laboratory, Argonne, IL, 60439, USA
[2]NOAA-Global Modelling Division, Boulder, CO 80305, USA
[3]Actalent Services, Chicago, Illinois 60606, USA

*Correspondence to*: Maria P. Cadeddu (mcadeddu@anl.gov)

**Abstract.** Boundary layer moisture variability at the Eastern North Atlantic (ENA) site is examined at monthly and daily time scales using 5 years of ground-based observations and output from European Center for Medium range Weather Forecast (ECMWF) reanalysis model. The annual cycle of the mixed layer water budgets is presented to estimate the relative contribution of large-scale advection, local moisture tendency, entrainment, and precipitation to balance the moistening due to surface latent heat flux on monthly timescales. Advection of colder and dry air from the North acts as an important moisture sink (~ 50% of the overall budget) during fall and winter driving the seasonality of the budget. Entrainment and precipitation contribute to the drying of the boundary layer (~25% and ~15% respectively) and the local change in moisture contributes to a small residual part. On a daily temporal scale, moist and dry mesoscale columns of vapor (~10 km) are analyzed during 10 selected days of precipitating stratocumulus clouds. Adjacent moist and dry columns present distinct mesoscale features that are strongly correlated with clouds and precipitation. Dry columns adjacent to moist columns have more frequent and stronger downdrafts immediately below the cloud base. Moist columns have more frequent updrafts, stronger cloud top cooling, higher liquid water path and precipitation compared to the dry columns. This study highlights the complex interaction between large-scale and local processes controlling the boundary layer moisture and the importance of vapor spatial distribution to support convection and precipitation.

## 1 Introduction

Marine boundary layer warm clouds cover vast areas of eastern subtropical oceans and persist for very long periods (Klein and Hartmann, 1993). These clouds reflect much greater amount of radiation back to space compared to the ocean surface, and hence are an important component of the Earth's radiation budget. It is challenging for Earth System Models (ESM) used for predicting the future climate to accurately simulate these clouds as they and the associated processes occur at much smaller spatial and temporal scales than the model resolution.

Marine boundary layer stratocumulus and shallow cumulus are maintained by boundary layer turbulence through the transport of water vapor above the lifting condensation level. Some cloud parameterizations use the moments of the joint



PDFs of temperature, water vapor and vertical air motion to simulate cloud properties. Hence changes in boundary layer

water vapor critically impact cloud properties. Further the marine boundary layer clouds exhibit a distinct mesoscale organization (Wood and Hartman, 2006) with scales of 20-50 km. Recent modeling studies have shown that cloud and rain properties are organized in mesoscale structures and are closely related to changes in boundary layer water vapor (Zhou and Bretherton, 2019). Several modeling studies highlight the role of "mesoscale humidity aggregation" and its positive feedback in amplifying the moisture variance, cloudiness, and precipitation (Bretherton and Blossey, 2017; Lamaakel and Matheou,

40  2022).

To address scientific issues related to marine low clouds, the Atmospheric Radiation Measurement (ARM) Climate research facility (Mather and Voyles, 2013) operates the heavily instrumented Eastern North Atlantic (ENA) site located in the Azores Archipelago. The location experiences a distinct annual and diurnal cycle in aerosols, clouds, precipitation, dynamic and thermodynamic fields (Zheng et al. Lamer et al. 2019; Ghate et al. 2021; Giangrande et al., 2019; Wu et al. 2020; Zheng

et al. 2018). It is evident from Figure 1 that that the water vapor mixing ratio at the site presents a well-defined annual cycle with lower mixing ratio in fall and winter and a higher mixing ratio in summer (July-August). The bulk of the vapor is located below 2 km, however profiles at the site appear perhaps moister in the free troposphere than what is typically found for example in the Southeast Pacific marine boundary layer (e. g. Bretherton et al., 2010). The free tropospheric humidity also exhibits an annual cycle, albeit much weaker than that of the mixed layer water vapor. The purpose of this work is

therefore to identify the relevant controlling factors (mesoscale and local) that influence the total moisture field in the boundary layer at the site and investigate how changes in the moisture field are connected to changes in clouds and precipitation.

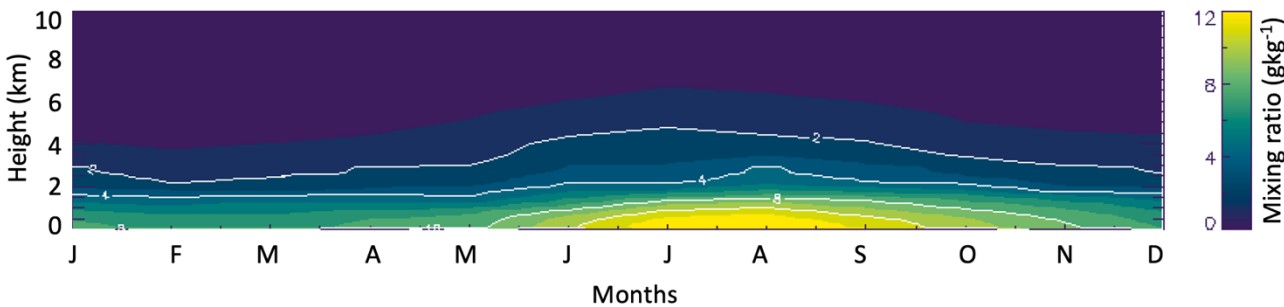

**Figure 1: Monthly averaged water vapor mixing ratio (shades) and contours (white lines) from radiosondes at the**

**ARM ENA site during marine conditions.**

We utilize the mixed layer framework that has traditionally been used to characterize the variability of boundary layer moisture and its controlling factors (e.g., Brost et al., 1992; Caldwell et al. 2005; Kalmus et al. 2014). In this framework it is assumed that the boundary layer is thermodynamically well-mixed and coupled to the surface with a constant profile of total

water (vapor + liquid) mixing ratio, and liquid water potential temperature. Although previous studied have shown a





prevalence of thermodynamically decoupled conditions over the open oceans (e.g., Wood and Bretherton, 2005; Serpetzoglou et al. 2006), the mixed layer framework offers a relatively straightforward way to characterize sources and sinks of the boundary layer thermodynamic variables. In the mixed layer framework, the boundary layer water field is modulated by advection, entrainment, precipitation, and surface latent heat flux. The validity of the mixed layer framework

65 has recently been shown to be sufficient to explain synoptic and monthly variability in the sub-cloud layer (Albright et al., 2022). In the first part of this work, we take advantage of excellent quality and continuous atmospheric measurements available at the ARM ENA site for the past 5 years (2015-2020) to characterize the boundary layer water vapor and its controlling factors through mixed layer water budgets. In the second part we use data collected during stratocumulus cloud conditions to diagnose the relationship of the mesoscale variability in the boundary layer water vapor with cloud,

70 precipitation, and radiation fields. The remainder of the paper is organized as follows: Section 2 presents an overview of the data and retrievals utilized with a discussion on the novelty of retrievals used to separate the contribution of cloud and drizzle to the total liquid water path. The annual cycle of precipitable water vapor (PWV) and liquid water path is discussed in Section 3. In Section 4 the boundary layer moisture budget is described in connection to the annual cycle of the mixing ratio shown in Fig. 1. Finally, the mesoscale variability of water vapor during 10 selected days is analysed in Section 5 and

75 the work is concluded with a summary and discussion section.

## 2 Instrumentation, Data and Retrievals

The Eastern North Atlantic is one of the ARM program's (Turner and Ellingson, 2016) permanent sites situated on the island of Graciosa (39.1°N, 28.0°W, 25 m) in the Archipelago of the Azores. The climate at the site is characterized by a wide

80 range of weather conditions influenced by the frequent arrival of midlatitude winter storms (e.g., Rémillard et al., 2012; Wood et al., 2015) and by the influence of trade winds. Recent studies have evidenced the connection between precipitation properties and large-scale conditions, for example an increased frequency of precipitating clouds in the wake of cold fronts (Lamer et al., 2020). The total cloud fraction at the site is higher in winter (Dong et al., 2014) while during summer the prevalence of a high-pressure system reduces cloud cover and promotes the prevalence of fair-weather conditions (Wood et

85 al., 2015).

### 2.1 Instrumentation

The site has several instruments to observe the aerosol, cloud, radiation, and thermodynamic fields. We mention here the main operating characteristics of the instruments used in the analysis. A vertically pointing ARM Ka-band Zenith Radar (KAZR) records the full Doppler spectrum and its first three moments at 2 second temporal and 30 m range resolution. The

90 KAZR was calibrated using a corner reflector and its accuracy is good within 3 dB (Kollias et al., 2019). A laser ceilometer operating at 905 nm wavelength records the profile of backscatter at 30 m range and 15 second temporal resolution along with the cloud base height. A Doppler Lidar operating at 1.5 $\mu$m wavelength records the backscatter and the mean Doppler velocity at 30 m range and 1 second temporal resolution. The ceilometer and the Doppler Lidar were calibrated by the

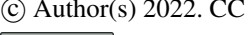



authors using the technique proposed by O'Connor et al. 2005. The details of the technique, as applied to this site, are

explained in Ghate et al., 2021. A Raman Lidar (RL), collocated to the KAZR, transmits at a wavelength of 355 nm and records the backscattered radiation at wavelengths of 355 nm, 387 nm and 408 nm, at 10 second temporal and 7.5 m range resolution. From this instrument profiles of water vapor mixing ratio are derived at 10-minute temporal resolution. Also present at the site is a 3-channel microwave radiometer (MWR3C) that measures the sky brightness temperatures at 23.8, 31.4 and 90 GHz. The MWR3C is automatically calibrated using tip curves as explained in Cadeddu et al., 2013. An

Atmospheric Emitted Radiance Interferometer (AERI) measures the sky brightness temperatures from 3 to 19 $\mu$ m wavelengths at 8-minute time resolution. Collocated with the remote sensors is a video disdrometer that measures surface rain rates at 1-minute resolution, and a surface meteorological station for surface temperature, humidity, winds, and pressure at 1-minute resolution. Radiosondes are launched at the site every 12 hours at 00 and 12 UTC, and measure profiles of temperature, humidity, pressure, and winds.

In addition to the instruments listed above, we also used output from the European Center for Medium Range Weather Forecasting (ECMWF, ERA5) reanalysis model over the region (Rodwell and Jung, 2020). Quantities from the ECMWF include hourly surface latent heat fluxes, large-scale subsidence, winds, and water vapor mixing ratio profiles. A summary of the instruments and retrievals used in the analysis is shown in Table 1.

## 2.2 Data selection

Five years and 4 months (August 1, 2015, to December 31, 2020) of data from all instruments in Table 1 were processed, except the RL, DL, and AERI that were used only in the 10 selected cases discussed in Section 5. Drizzle liquid water content below cloud base, as well as retrievals of water vapor, cloud, and drizzle water path were derived every minute and hourly averaged to match the time resolution of the ECMWF data. After the necessary quantities were derived cases that were classified as "only marine conditions" were identified and used for further analysis. Marine conditions were defined,

following Ghate et al., 2021, by selecting data corresponding to surface wind direction (measured clockwise) greater than 310° or less than 90°, thereby eliminating cases where the boundary layer may have been influenced by the island itself. Out of 52,608 total observations, 15,972 hours (30%) were identified as marine conditions.

Total, cloud, and drizzle water paths are produced when precipitation does not reach the surface or when it is not enough to contaminate the measurements. The mean disdrometer rain rate at the surface, when the retrievals converged, was less than

0.05 mm hr$^{-1}$. The retrievals converged in 12,131 hours (76% of the marine cases). This limitation in the data selection due to the microwave radiometer's inability to produce reliable measurements during heavy precipitation reduces the data sample size, but it doesn't alter the climatological features of the dataset.

## 2.3 Retrievals

The KAZR and ceilometer data were combined to retrieve profiles of drizzle properties below the cloud base using the

technique proposed by O'Connor et al., 2005. These retrievals were further combined with the brightness temperatures from



the MWR3C to obtain water vapor, cloud and drizzle water path using the Synergistic Passive and Active Retrieval of Cloud Properties (SPARCL) (Cadeddu et al., 2017, Cadeddu et al., 2020).

| Instrument | Retrieved Physical quantity | Used for |
|---|---|---|
| Surface meteorological station (MET) | Surface wind direction | Identification of marine cases |
| Radiosondes | Boundary Layer thermodynamic properties and depth | TROPO-OE and SPARCL retrievals; Total Water and Mass Budget |
| Disdrometer | Rain rate and surface precipitation flux | Total Water Budget analysis |
| Ka-band ARM Zenith Radar (KAZR) and Ceilometer | Drizzle properties below cloud base | Separation of cloud and drizzle water path; Estimation of $q_t$ in moist budget and in MSE computations. |
| 3-channel microwave radiometer (MWR3C) | Precipitable Water Vapor (PWV), Cloud Water Path (CWP), Drizzle Water Path (DWP), Liquid Water Path (LWP) | Climatology, Total water budget analysis, Mesoscale humidity analysis, and cloud adiabaticity |
| European Center for Medium-range Weather Forecasting reanalysis model (ERA5) | Surface turbulent fluxes, large-scale subsidence, and winds | Total water and Mass Budget analysis. |
| Raman lidar (RL) and Atmospheric Emitted Radiance Interferometer (AERI) | 10-minute profiles of water vapor mixing ratio. | TROPO-OE, Mesoscale humidity analysis. |
| Doppler Lidar (DL) | 1-minute profiles of vertical velocity | Mesoscale humidity analysis. |

**Table 1: Instruments used in the analysis, physical quantities, and use of data.**





Uncertainties in the water vapor and liquid water path retrievals are of the order of 0.5 mm for water vapor and about 15 g m$^{-2}$ for liquid water path. For ten selected cases of weakly precipitating marine stratocumulus clouds vertical profiles of water

vapor mixing ratio were derived every 10 minutes using the optimal estimation retrieval TROPoe (formerly AERIoe, Turner and Löhnert, 2014, Turner and Blumberg 2019, Turner and Löhnert, 2021). These retrievals use combined data from the Raman lidar, AERI, and microwave radiometer and have an average uncertainty of 0.6 g kg$^{-1}$ in the first 3 km. They are further discussed in Section 5.

The retrieval of LWP with SPARCL enables the separation of cloud and drizzle water path by exploiting the different

signature of cloud and drizzle drops on the 90 GHz frequency of the microwave radiometer. Because the retrieved column integrated values of liquid water and water vapor form the basis of this work, we first compare SPARCL retrievals to the traditional LWP product (MWRRET, Turner et al., 2007) available in the ARM Archive. Traditionally the total liquid water path retrieved by radiometers is assumed to represent the cloud water path, however in the presence of drizzle, cloud liquid water is only a component of the total liquid water path, and the radiometric retrievals cannot distinguish between the cloud

and the below-cloud drizzle part. In the presence of drizzle drops with diameter larger than ~100 $\mu$m, the brightness temperature at 90 GHz is affected by Mie scattering effects that, if not interpreted properly, can result in an overestimation of the LWP. These aspects of the retrievals are shown in Figure 2. In the following discussion only boundary layer clouds with cloud fraction from the ceilometer greater than 0.99 were selected (total of 3580 hours).

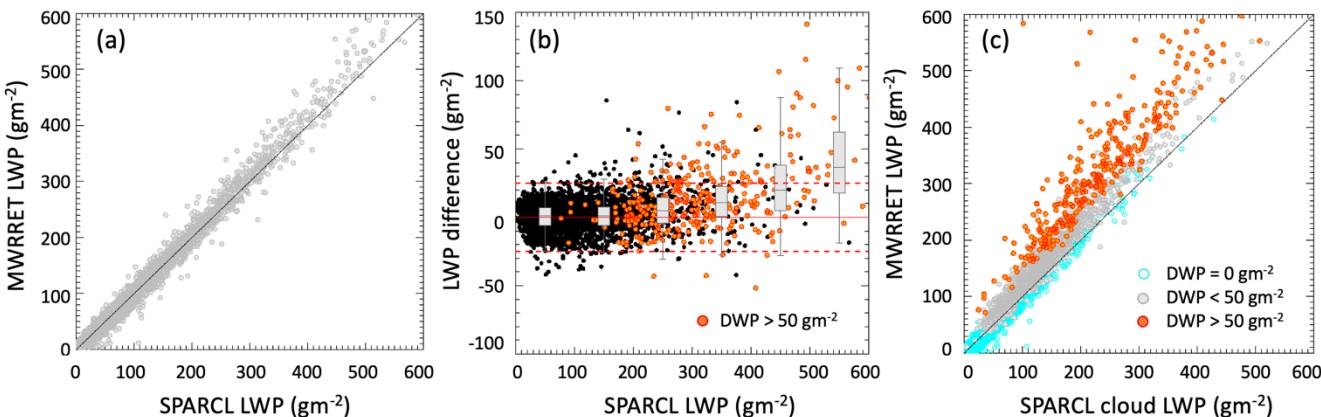

**Figure 2: (a) Scatterplot between total LWP in this work (SPARCL) and physical retrieval (MWRRET) LWP. (b) Difference between the two retrievals (MWRRET minus SPARCL) for all samples (black) and samples with below cloud drizzle water path greater than 50 g m$^{-2}$ (orange). The orange dashed horizontal lines indicate ± 25 g m$^{-2}$. (c) Scatterplot between SPARCL cloud water path and MWRRET LWP. Colors represent cases with no drizzle (cyan), drizzle water path below cloud base less than 50 g m$^{-2}$ (grey), and drizzle water path below cloud base greater than 50**

**g m$^{-2}$.**





As evident from Fig. 2a the new LWP compares very well with the traditional optimal estimation retrieval (MWRRET) available in the ARM Archive (Turner et al., 2007, Cadeddu et al., 2013). Looking closer, the subtle differences between the two retrievals are apparent when the LWP is higher than 300 g m$^{-2}$. This is better visualized in Fig. 2b where differences

between the retrievals are displayed as a function of LWP. When the drizzle water path below cloud is higher than 50 g m$^{-2}$, MWRRET (which assumes all the hydrometeors are in the Rayleigh scattering regime) overestimates LWP because of neglect of scattering effects from larger hydrometeors. As shown in Fig. 2c, even in the presence of light precipitation, the amount of drizzle water in clouds is non-negligible and the total liquid water path coincides with cloud water path only in non-drizzling clouds (cyan points in Fig. 2c). When drizzle is formed however, attributing the entire column to cloud drops

leads to an incorrect interpretation of the data. It should be noted that the SPARCL retrieval algorithm also derives in-cloud drizzle water path and hence the total drizzle water path (cloud and below cloud) is greater than that derived from the retrievals using data from active sensors such as the radars and lidars.

## 3 Annual cycle and adiabaticity


The annual cycle of water vapor and liquid water is characterized next. Consistent with Fig. 1 precipitable water vapor shows a distinct annual cycle (Fig. 3 a, b) characterized by higher average water vapor in summer (2.99 cm in August) and dryer conditions in late fall and winter (1.4 cm in March). The radiometric retrievals are in good agreement with the radiosondes in both magnitude and variability. The annual cycle is mostly visible in the lower troposphere (below 3 km) but

is still present although much weaker in the mid troposphere above 3 km (pink boxes). The proportion of free tropospheric PWV to the total amount ranges from 14% in February to 20% in June. Therefore, the annual cycle doesn't have the same amplitude in the upper and lower troposphere resulting in a stronger contribution of the free troposphere to the total PWV in summer compared to winter. This points towards greater contribution of mid-tropospheric vapor towards reducing the boundary layer radiative cooling in the summer compared to the winter months.

The total LWP during marine conditions (Fig. 3c) is characterized by a weak seasonal cycle with higher LWP and higher variability during the fall and winter months. When summarized in a seasonal cycle, the mean LWP (in g m$^{-2}$) is 82.3, 64.6, 67.3, 75.1 (DJF, MAM, JJA, SON). For these weakly precipitating clouds captured by the radiometer, the monthly mean LWP is less than 100 gm$^{-2}$ throughout the year with maximum hourly values not exceeding 300 gm$^{-2}$. The cloud component, shown in light brown in Fig. 3c, constitutes the bulk of the total LWP (in dark brown) and it mimics the total LWP. Drizzle

water path, which includes both, below- and in-cloud components, and is shown in Fig. 3d, is detectable through the entire year, even when the average LWP is low. Drizzle water path displays a pronounced seasonal cycle with averages in fall and winter about 17% higher than in summer and spring. The ratio of drizzle-to-total water path increases from ~ 11% in summer to ~15-17 % in winter. This propensity of clouds with similar total LWP to produce more drizzle in winter and fall may relate to a change in the prevalent typology of clouds and to the occurrence of midlatitude winter storms in fall and winter. It

points out to the fact that liquid water path is only one of the factors controlling drizzle production. A prominent feature of





the annual cycle at the site is the weak anticorrelation between the water vapor and the liquid water path. This feature is discussed in more detail in the next section where the relative contribution of the processes affecting the boundary layer moisture fluxes are examined.

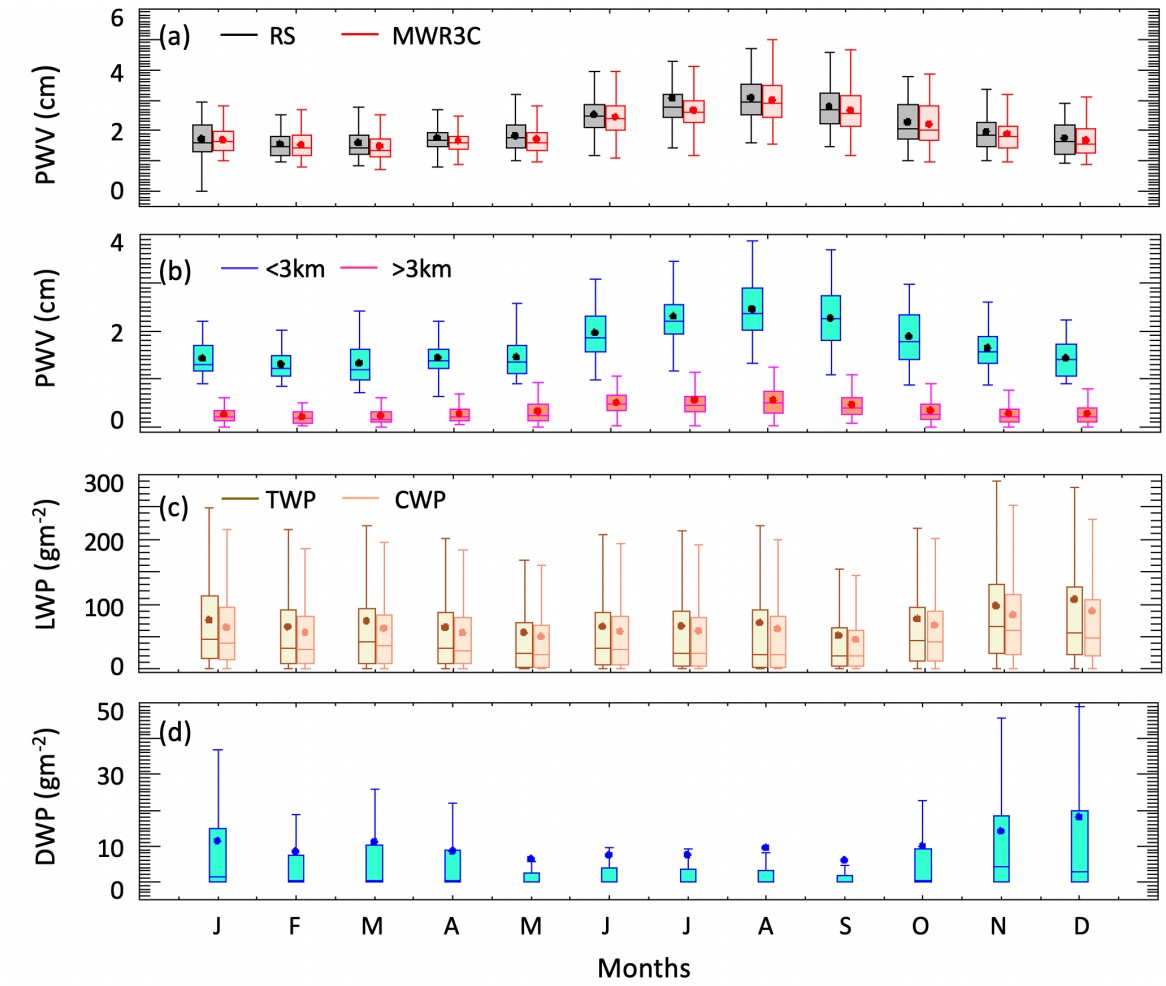


**Figure 3: a) Annual cycle of total PWV from radiosondes (RS) in grey and from the microwave radiometer in red; b) Monthly mean of PWV below 3 km in cyan and above 3 km in pink from radiosondes; c) Monthly mean total (dark brown) and cloud (light brown) water path derived from the MWR3C; d) Monthly mean drizzle water path derived from the MWR3C. The dots denote the means, the boxes enclose the interquartile range (IQR), and whiskers extend**

**to 1.5 times the IQR.**





## 3.1 Cloud adiabaticity

The large dataset gives us an opportunity to analyze cloud adiabaticity at the site by estimating the ratio between the retrieved liquid water path and the liquid water path calculated using the adiabatic assumption for cases coincident to
radiosondes (Zuidema et al., 2005, Albrecht et al., 1990). The quantities necessary for the computation of the adiabatic LWP, such as cloud top height, temperature, pressure, and humidity profiles, are taken from radiosondes and the cloud base height is taken from the ceilometer. A total of 304 points satisfied all the necessary requirements for the comparison (absent or weak precipitation, marine condition, coincident radiosonde, cloud fraction > 0.99) and are shown in Fig. 4 and in Table S1 in the supplemental material.

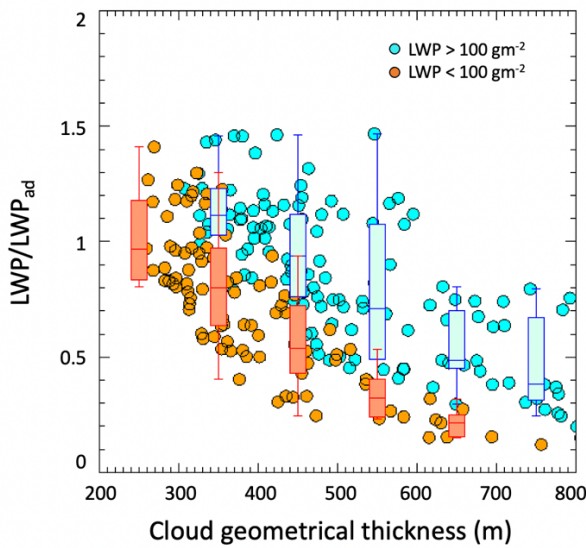

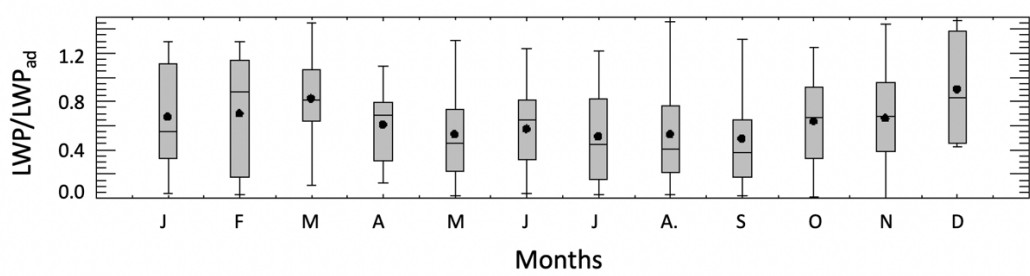

**Figure 4: Top: Adiabaticity of sampled clouds against geometrical thickness for cloud with 50 <LWP < 100 g/m$^2$ (orange) and >100 g m$^{-2}$ (cyan). Bottom: Annual cycle of adiabaticity.**



Clouds appear to be increasingly sub-adiabatic with increasing cloud geometrical thickness. These results are consistent with previous observations that evidenced the mostly sub-adiabatic nature of marine clouds (e.g., Min et al., 2012 over the south pacific and Zhou et al., 2019 at the ENA site). In this dataset precipitation doesn't seem to influence the degree of subadiabaticity leading to the speculation that entrainment of dry upper tropospheric air may be the main reason for the departure from adiabatic behavior. Uncertainties in the estimation of the cloud boundaries and liquid water path are likely

responsible for the small percentage of clouds that result super-adiabatic. Clouds are on average slightly more adiabatic and display higher variability in winter than in summer (Fig. 4, bottom). This could be due to deeper boundary layers, stronger turbulence and higher prevalence of thermodynamically coupled boundary layers during the winter months as compared to the summer months (Wang et al. 2021).

**4 ENA Regional moisture budget**

The 5 years of data, together with output from ERA5 model are used to estimate the boundary layer moisture budget at the ENA. The observations are used to investigate which mesoscale processes control the moisture variability at the site on monthly and seasonal temporal scales. For the estimation of the moisture budget, we use the total mixing ratio $q_t = q_v + q_L$,

which is the sum of water vapor mixing ratio ($q_v$) and liquid mixing ratio ($q_l$) when clouds are present and follow the formalism of Caldwell et al., 2005. We therefore express the boundary-layer averaged formulation (in units of W m$^{-2}$):

$$\frac{\mathcal{L}}{g}\hat{p}\left(\frac{\partial \langle q_t \rangle}{\partial t} + \langle \boldsymbol{v} \cdot \boldsymbol{\nabla} q_t \rangle\right) - LHF - \mathcal{L}\,P - \frac{\mathcal{L}}{g}\widehat{\omega}_e \Delta q_t = 0 \qquad (1)$$

Where $\mathcal{L}$ and $g$ are the latent heat of vaporization (2.5 X 10$^6$ J kg$^{-1}$) and the acceleration due to gravity (9.8 m s$^{-2}$), brackets $<$

$>$ indicate averages between the surface and the top of the boundary layer, and $\hat{p} = p_0 - p_t$ is the difference between surface pressure and pressure at the top of the boundary layer. The first 2 terms represent the time change in the PBL-averaged moisture (local tendency) and the large-scale horizontal advection to the region with $\boldsymbol{v}$ being the wind vectors and $\boldsymbol{\nabla} q_t$ the horizontal gradient. The third and 4$^{th}$ terms represent the surface latent heat fluxes (LHF) as reported by ERA5, and the precipitation fluxes obtained from the disdrometer rain rate ($P$) expressed in kg m$^{-2}$ s$^{-1}$ and multiplied by the latent heat of

vaporization ($\mathcal{L}$). The last term in eq. (1) is related to the turbulent fluxes due to entrainment of dry air at the top of the boundary layer. Specifically, $\widehat{\omega_e}$ is the entrainment velocity and $\Delta q_t$ the gradient of total water mixing ratio across the top of the boundary layer. Some quantities necessary in Eq. (1) are not available from measurements and are therefore calculated using ECMWF reanalysis data as shown in Table 1. Below we provide an overview of the data and details on the methodology used to calculate each term. All the terms were estimated on an hourly basis and averaged each month. All

components were screened for outliers eliminating points beyond 2 standard deviations from the monthly mean and were passed through a 24hr running average. The vertically gridded data (radiosondes and ECMWF profiles) were interpolated on



a common vertical grid of 50 m vertical resolution. Values for the quantities in (1) are shown in units of W m$^{-2}$ in all the subsequent analysis.

## 4.1 Datasets used for each term

The planetary boundary layer (PBL) height is used to determine the highest limit for integrating the water vapor profiles and to calculate the entrainment rate. The ARM Value Added Product (VAP) available from the ARM Archive (Sivaraman et al., 2013) reports PBL heights from the radiosondes based on the Heffter (1980) method that identifies the base of the inversion from the gradient of the potential temperature. Because the ENA site only launches two radiosondes per day (at 00 and 12 UTC), we assume the PBL height to be as reported by the radiosonde within ±6 hour of each radiosonde. If PBL height data

were not available for one entire day the day was flagged as missing.

The local change $\frac{\partial \langle q_t \rangle}{\partial t}$ in the moisture profile at the site was estimated as follows: Vertical profiles of $q_v$ are available from radiosondes twice a day. The vertical distribution of humidity between 0-12 UTC (12-24 UTC) was kept constant and equal to the morning (evening) radiosonde reported values. The profiles were scaled to the MWR3C-retrieved hourly PWV using a height-independent scale factor (e.g., Turner et al. 2003). For the estimation of $\boldsymbol{q}_L$, the hourly averaged cloud liquid water

path from the SPARCL retrieval was distributed adiabatically between cloud base and cloud top. Because the PBL height is only available twice per day or less, monthly profiles of PBL heights were used to identify the top of the boundary layer. The errors introduced by this approximation are small because of the limited variability of the PBL height, as shown later. Profiles of $\boldsymbol{q}_t = \mathbf{q}_v + \mathbf{q}_L$ were then averaged between the surface and the top of the boundary layer, the difference between successive hours calculated and multiplied by $\hat{p}$.

The large-scale moisture advection $\langle \boldsymbol{v} \cdot \boldsymbol{\nabla} q_t \rangle$ was estimated from ERA5 data by calculating the horizontal gradient of the moisture field across grid points adjacent to the ENA site using the equation:

$$\boldsymbol{v} \cdot \boldsymbol{\nabla} q_t = v_{NS} \frac{\partial \boldsymbol{q}_{NS}}{\partial \boldsymbol{x}_{NS}} + u_{EW} \frac{\partial \boldsymbol{q}_{EW}}{\partial \boldsymbol{x}_{EW}} \qquad (2)$$

Where $\boldsymbol{v}_{NS}$ and $\boldsymbol{u}_{EW}$ are the boundary layer averaged meridional and zonal components of the wind, and gradients denote

differences $\boldsymbol{q}_{XY} = \boldsymbol{q}_Y - \boldsymbol{q}_X$ between grid points (0.25 X 0.25 degrees) adjacent to the ENA site. The bold notation indicates that the differences are calculated at all vertical points before averaging over the boundary layer. Note that only the vapor component was estimated, and the contribution of liquid water advection was neglected. As we selected only marine conditions for our analysis, we expect this component to be dominated by a prevalence of colder and drier air from the north. Hourly values of surface latent heat flux from the ERA5 reanalysis model data are used for this analysis. Precipitation data

from the disdrometer at the site were hourly averaged and passed through a 24-hour running average to eliminate excessive noise. In addition, the highest 5% values that yield extremely high precipitation fluxes were eliminated. As an example, the





mean flux contribution due to the highest 5% of the data is 365 [185-3470] Wm$^{-2}$ compared to a mean value of 16 [0 – 184] Wm$^{-2}$ due to the remaining 95% of the population. The last term of the budget equation (1) requires the evaluation of the moisture gradient at the top of the boundary layer and of the entrainment rate. The term $\Delta q_t$ was defined as the difference

between the total mixing ratio above the boundary layer inversion and the PBL-averaged $q_t$ and was therefore estimated from radiosondes, retrieved water vapor and cloud liquid water path. The annual cycle of all the quantities described above is shown in Figure 5.

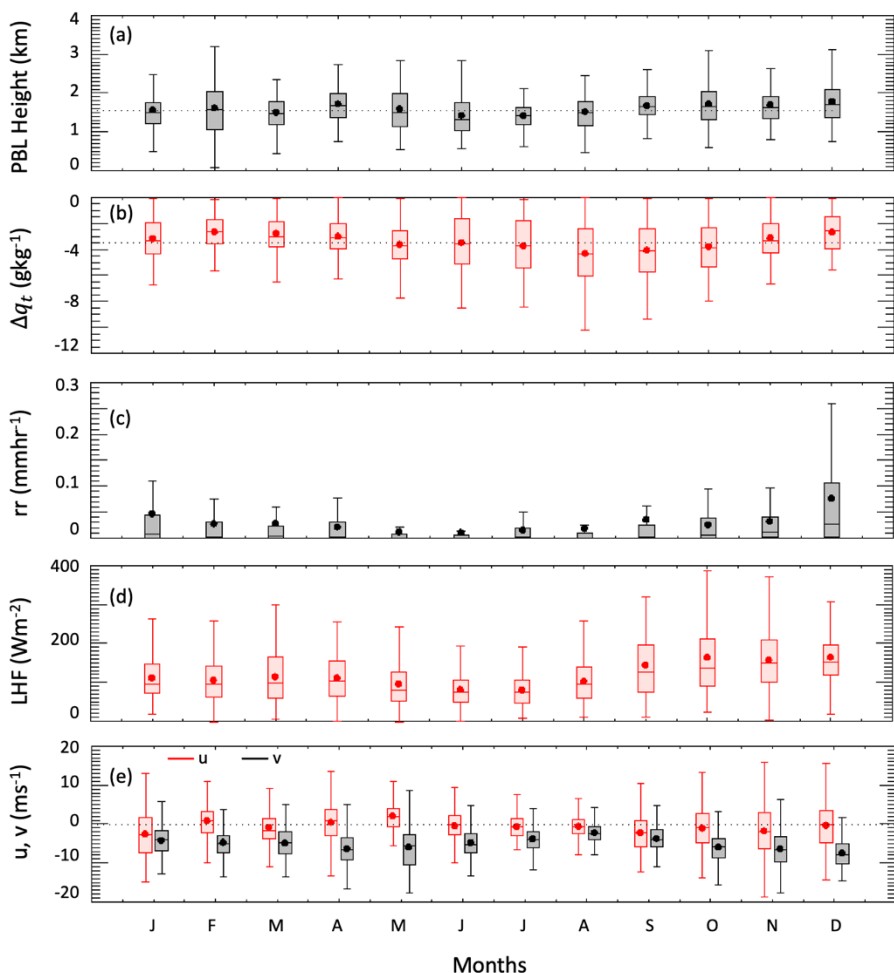

**Figure 5: Monthly box-whisker values of (a) PBL height, (b) humidity gradient at the boundary layer top, (c) rain rate, (d) latent heat fluxes, and (e) zonal (u) and meridional (v) components of the wind. Dotted lines represent the annual mean in (a) and (b) and the zero line in (e).**





The monthly mean PBL height didn't display a marked annual cycle, but rather small perturbations about an annual mean

value of 1.6 ± 0.5 km. On average a deeper boundary layer (PBLH=1.7 km) is found in fall (SOND) and a shallower boundary layer (1.4 km) is found in June and July. During summer months the humidity gradient at the PBL top was strongest and exhibited greatest variability. Conversely, a deeper average boundary layer in fall and winter season had weaker humidity gradients at the PBL top. The southward boundary layer winds were stronger in the winter months compared to the summer months, and so were the surface LHF fields and rain rates. Collectively this figure suggests that

winter months exhibit on average a deeper boundary layer, weaker boundary layer inversion, higher winds, fluxes, and rain rates compared to the summer months. This is consistent with the finding in Ghate et al., 2021 who observed higher boundary layer turbulence in winter than in summer.

### 4.2 Boundary layer mass budget and entrainment rates


Assuming a near-constant air density within the boundary layer, the mass budget can be written in terms of the changes in PBL height. Here we utilize the boundary layer mass budget to calculate the entrainment rates on monthly timescales. The entrainment velocity at boundary layer top is balanced by local change in the boundary layer, advection of the PBL top, and largescale vertical air motion at the PBL top (equation 3).


$$\omega_e = \frac{\partial \hat{p}}{\partial t} + \boldsymbol{v} \cdot \boldsymbol{\nabla}\hat{p} - \hat{\omega}_s \tag{3}$$

Following Wood and Bretherton (2006) all terms were monthly averaged before using in (3) and the monthly averages of the terms were additionally filtered to eliminate outliers and unphysical values and to reduce the large noise of the dataset. The term $\frac{\partial \hat{p}}{\partial t}$ was computed taking the difference between two successive soundings (12-hour time step). Advection of the PBL

top $\boldsymbol{v} \cdot \boldsymbol{\nabla}\hat{p}$ and the largescale vertical air motion were estimated from ERA5. Although the mass budget was computed in pressure unit (Pa s⁻¹) for the calculations of the fluxes in eq. (1), we convert it to mm s⁻¹ for the figures and in the following discussion to facilitate comparison with previous work.

The monthly mean components of the mass budget and resulting entrainment rates are reported in Table S2 in the supplemental material. The annual mean local tendency of the PBL top, advection, and largescale subsidence at PBL top

were -0.0061, 0.012, 0.036 Pa s⁻¹ which translates approximately into 0.6, -1.0, and -3.2 mm s⁻¹ respectively. The values of the local tendency of the PBL top and advection were small during all months and the advection of the boundary layer height was less than 40% of the entrainment rate in winter but was somewhat more prominent in summer. The small contributions of the local tendency and of the advection term resulted in a near balance between entrainment velocity and subsidence (Fig. 6). Entrainment rates exceeded subsidence of ~20% in January, and for the rest of the months entrainment balanced or was

slightly weaker than subsidence. However, the high standard deviation indicates a large variability throughout the year. The




near balance of entrainment and large-scale turbulence is consistent with the observed small variability of the PBL height. On the average the entrainment rates found here are slightly lower that what reported in Wood and Bretherton (2004) along the Pacific coast, the variability is however well in the range of values found in previous studies. The calculated entrainment rates ($\omega_e$) display a weak annual cycle with higher values in fall and winter and lower values in summer with averages

varying from 1.7 mm s$^{-1}$ in July to 4.1 mm s$^{-1}$ in January. The annual cycle is consistent with higher turbulence during the winter months compared to the summer months. These values are also in broad agreement with those found in previous studies at different locations (e. g. Painemal et al., 2017; Ghate et al. 2019; Albrecht et al., 2016) for stratocumulus clouds.

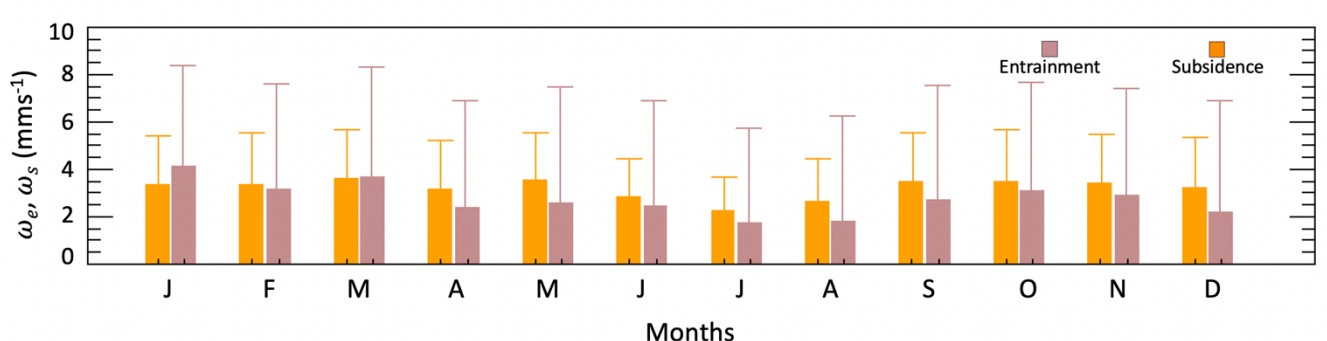

**Figure 6: Monthly entrainment rates $\omega_e$ and subsidence rates $\omega_s$ with associated 1 standard deviation.**


**4.3 Monthly and seasonal moisture budget**

With all the terms in equation (1) accounted for, the annual cycle of the moisture budget is shown in Figure 7. Because the focus is on marine conditions, the large-scale advection is predominantly from the North and hence represents a boundary

layer drying along with entrainment and precipitation. The local tendency term is often assumed to be zero (e.g., Caldwell et al., 2005 and Kalmus et al., 2014), but here is calculated explicitly for each month. The term is small and, except for the months of January and May, acted as a weak moisture sink. Annually the surface latent heat flux, which constitutes the only moisture source, is balanced by the local change, advection, entrainment, and precipitation with a residual term of 9.2 Wm$^{-2}$. The fluxes are highest in the winter months and lowest in the summer months, anti-correlated to the boundary layer vapor

annual cycle shown in Figure 1 and 3 (a,b). This largely represents the meteorological correlation of calmer winds, lower sea-air temperature difference, shallower boundary layers, and weaker advection with increased vapor loading during the summer months compared to the winter months.

The seasonality of the large-scale advection term is also the factor that determines the seasonality of the overall budget. The contribution from advection in October and December is almost twice the contribution in June and July. Corresponding to

the advection of colder and dry air from the north, latent heat fluxes increase from September to December. In November and December, the monthly vapor source and sink budget are not entirely balanced with residual as high as 50 W m$^{-2}$ in





November. To better summarize the dataset, Fig 7b and Table 2 show the seasonal values of all terms and the residuals. Seasonally the positive (LHF) and negative terms of the moisture budget balance in within ±20 Wm$^{-2}$. As shown in Table 2 there are large standard deviations associated with the single components of the budget. These standard deviations are

partially a result of the uncertainty associated to the computations of the terms (model, measurements, retrievals, approximations, etc.) and are also due to the broad range of atmospheric conditions encountered in the dataset used. The budget closure is well within the estimated uncertainty. Annually averaged, the moistening from the surface is primarily balanced by advection drying (~50%) and entrainment drying (~25%) with the rest compensated through precipitation removal. Overall, the budget analysis highlights the complexity of the moisture variability in the region as advection is

determined by largescale features external to the boundary layer, while the entrainment and precipitation are determined by boundary layer internal properties. This issue is discussed further in the last section.

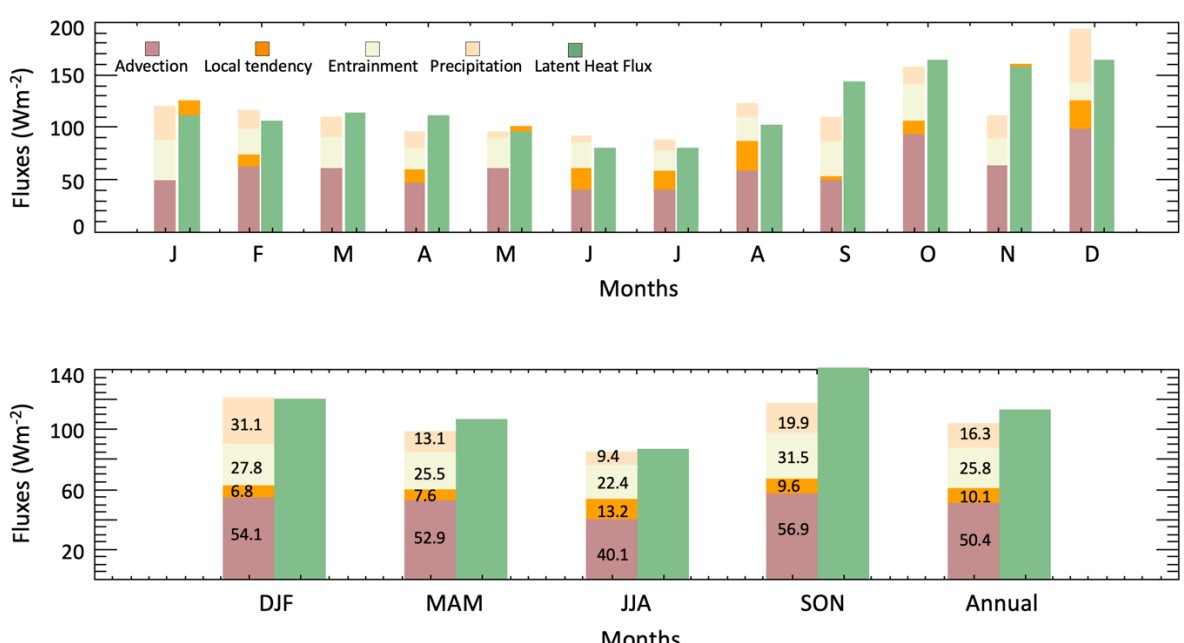

**Figure 7: Monthly a) and seasonal b) components of the moisture budget. Left bars are moisture sinks, right bars are**
**moisture sources. Color bars represent advection (brown), local tendency (orange), entrainment (cream), precipitation (pink), and latent heat fluxes (green). Numbers in the bottom plot are percent values. The residual terms are not shown in this figure.**






| | DJF | MAM | JJA | SON | YEAR |
|---|---|---|---|---|---|
| $\dfrac{\mathcal{L}}{g}\hat{p}\dfrac{\partial\langle q_t\rangle}{\partial t}$ | -8.4± 67.5 (6.8%) | -7.6±62.2 (7.7%) | -13.2±58.5 (15.5%) | -9.6±66.5 (8.1%) | -10.1±63.5 (9.7%) |
| $\dfrac{\mathcal{L}}{g}\hat{p}\langle\boldsymbol{v}\cdot\boldsymbol{\nabla}q_t\rangle$ | -54.1±51.3 (44.6%) | -51.9±47.5 (52.9%) | -40.1±41.6 (47.1%) | -56.9±51.2 (48.3%) | -50.4±48.2 (48.7%) |
| $\dfrac{1}{g}\omega_e\Delta q_t$ | -27.8±14.1 (22.9%) | -25.5±12.1 (25.4%) | -22.4±12.5 (26.3%) | -31.5±15.9 (26.7%) | -25.8±14.3 (23.7%) |
| $\mathcal{L}\,P$ | -31.1±56.2 (25.6%) | -13.1±24.7 (13.1%) | -9.4±22.4 (11.1%) | -19.9±32.9 (16.9%) | -16.3±31.6 (15.7%) |
| E | 119.7±57.7 | 106.7±62.2 | 87.1±48.1 | 141.2±68.5 | 112.7±63.3 |
| Residual | -1.7 | 8.6 | 2.0 | 23.3 | 9.2 |

**Table 2: Seasonal average values and standard deviation of the budget components. In parenthesis are the contributions of each negative term to the total boundary layer drying. Residuals are computed as the difference**
**between source and sinks (Wm$^{-2}$).**

## 5 Mesoscale variability of water vapor

Now we focus on shorter time scales (days and hours) to understand how mesoscale perturbations of water vapor locally
affect boundary layer radiative cooling, clouds, turbulence, and precipitation. Mesoscale organizations of clouds are visible
from satellites and have been extensively studied to understand how they affect or are affected by water vapor and
precipitation (Stevens et al. 2019; Wood and Hartmann, 2006; Bretherton and Blossey, 2017). Mesoscale self-aggregation is
however not easily discernible with ground-based observations and some attempts have been made at gathering
observational evidence from satellite (Lebsock et al., 2017) and from specific sites and reanalysis (Shultz and Stevens,
2018). Here we focus on the spatial organization of water vapor on a scale of 10 km by identifying its perturbations over a
background state similarly to the framework recently developed by Zhou and Bretherton (2019).

### 5.1 Hourly spatial and temporal distribution of moisture

To analyze the mesoscale (10 km) distribution of water vapor we use 1-minute averaged PWV and LWP from the
radiometer, and rain rate from the disdrometer, during 10 selected days at the ENA between 2016 and 2019. A list of





selected cases is in the supplemental material, Table S3. The selected days displayed persistent boundary layer cloudiness
and at times precipitation. We also use vertical profiles of mixing ratio below cloud base from combined Raman lidar, AERI,
and MWR (see TROPO-oe in Table I) produced every 10 minutes and then interpolated over 1 minute. When compared to
radiosondes at the site the coincident TROPO-oe retrievals well reproduce the mean profile. The bias between radiosondes
and retrievals is less than 0.35 g kg$^{-1}$ (or less than 6%) from 150 m to the cloud base and the standard deviation of the
differences is about 0.1 g kg$^{-1}$ (Figure S1). Hence the retrieved vertical profiles of mixing ratio are well suited to investigate
the mesoscale variability of the moisture field.

The background water vapor value is found by implementing a running average window with a width of 100 km on the
PWV, and a (sub-mesoscale) perturbed field is found by implementing a running average window with a width of 10 km.
Because the data are on a uniform one-minute time grid (=60 s) the number of minutes over which the averages are
performed is determined through the daily average wind speed at cloud base. For example, if the average wind speed is 7 m
s$^{-1}$ the full mesoscale window is obtained by averaging over 238 minutes. The background field is subtracted from the
perturbed field to identify the magnitude of the mesoscale perturbations. Mesoscale regions with positive difference are
"moist" cells, those with negative difference are "dry" cells. The statistics of the 345 identified perturbations is shown in
Fig. 8.

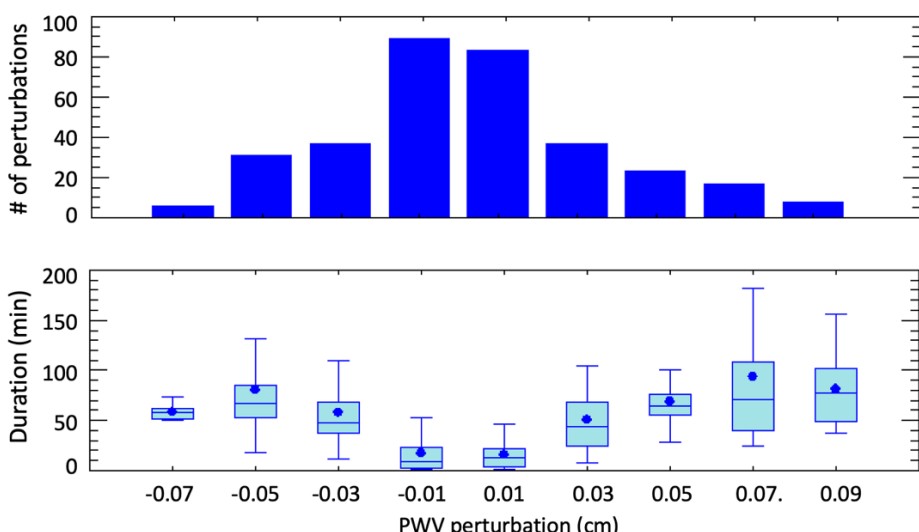

**Figure 8: Top: Distribution of the 345 water vapor perturbations found in the 10 days analyzed. Bottom: Duration of
water vapor perturbations binned by the amplitude of the perturbation, N= 6, 31, 37, 89, 83, 37, 23,17, 8 for a total of
331 points. The 14 missing cases are distributed in the bins below -0.07 cm and above 0.09 cm with less than 5 points
in each bin.**





About half (57%) of the mesoscale perturbations are weak and of short duration (~25 min); 37% of the perturbations are of medium magnitude (±0.05 cm) and about 6% of them can be classified as strong (±0.1 cm). Positive and negative

perturbations balance each other in strength with a slight prevalence of positive perturbations for the strongest cases as shown later. The duration of the perturbations is proportional to their strength. Medium and strong perturbations last on average 68 and 129 minutes (1.1 and 2.15 hours) respectively indicating a marked difference in the wind speed at the cloud base.

Examples of a case with strong and weak mesoscale organization are shown in Figure 9. Day 2019/04/04 was characterized

by intermittent precipitation and areas of higher localized moisture in regions of precipitating shafts. The vertical distribution of the mixing ratio perturbations from the background state, clearly shows a spatial structure where columns of moist and dry cells alternate over the site. Moist regions are located below the cloud base and inside the precipitating shafts. The moist (dry) columns correspond to regions of increased (decreased) liquid water path and precipitation respectively. In the absence of precipitation (as for example in day 2019/06/08 towards the end) mesoscale perturbations from the background state are

absent and the spatial structure of the mixing ratio perturbations is less defined.

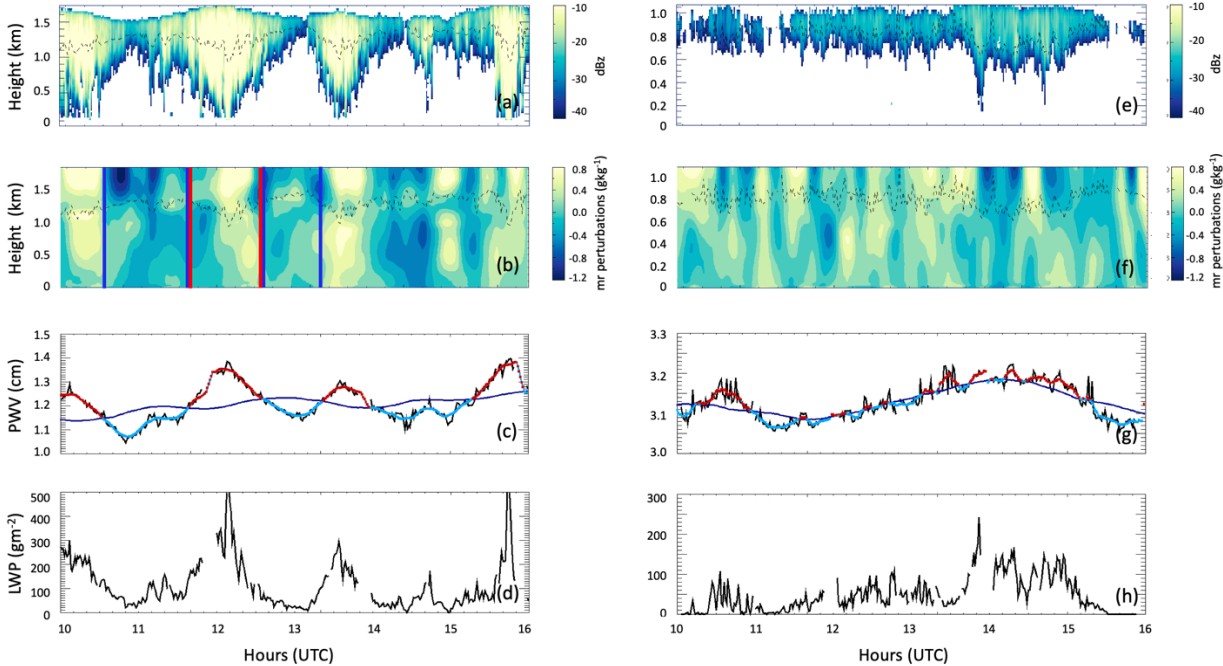

**Figure 9: From the top: radar backscatter, mixing ratio perturbations, PWV, and liquid water path on 2019/04/04 (left) and 2019/06/08 (right). The blue and red vertical bars on panel (b) represent examples of the beginning and the**

**end of dry (blue) and moist (red) adjacent columns. The dark blue line in panel (c) is the background state and red and cyan segments mark moist and dry perturbations.**





Moist and dry neighboring columns lasting longer than 10 minutes for a total of 143 columns were identified and the average properties of each cell were compared to those of its preceding and following neighbors. The turbulence forcing (cloud top
cooling, surface fluxes, etc.) varied greatly between the cases and hence we choose to characterize the differences between neighboring columns rather than aggregating solely based on the water vapor perturbations as was done in Zhou and Bretherton, 2019. Our results are however consistent with theirs.

Fig. 10 shows that dry and moist columns are preceded and followed by perturbations of opposite sign and similar amplitude. The exception are cases in which there are strong positive perturbations represented by the last 2 bins on the right
(PWV perturbations > 0.05 cm). In this case the dry neighbors on each side of the moist cell are weaker and present a weak asymmetry between the preceding and the following dry columns, with the preceding column being somewhat dryer than the following column.

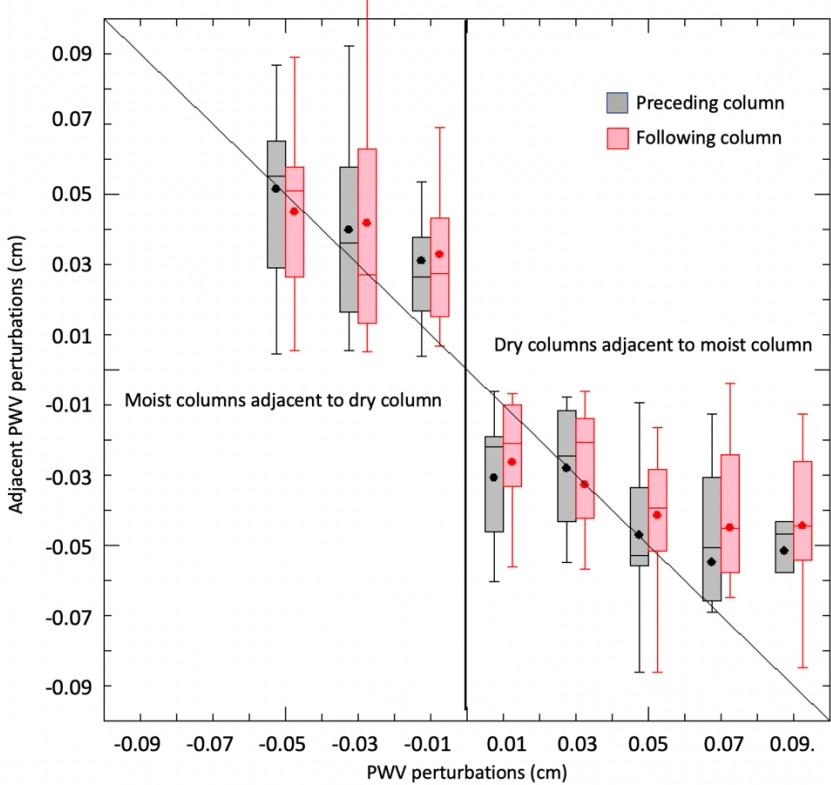

**Figure 10: Average water vapor perturbations in the columns preceding (black) and following (red) a central**
**perturbation identified in the x-axis. The diagonal line indicates the point where vapor perturbations in adjacent columns, have the same magnitude as the central column, N=25, 22, 19, 15, 20, 15,13, 5 for a total of 134 points. The 9 missing cases are distributed in the bins below -0.05 cm and above 0.09 cm with less than 5 points in each bin.**



We calculated cloud top radiative cooling, rain rate, cloud, and drizzle water path in each dry and moist mesoscale column.
These are shown in Fig. 11 as *differences* between the central and the two adjacent columns. Moist patches always have strong (more negative) cloud top cooling compared to the adjacent dry regions, even when the water vapor perturbation is weak, but a dependence on the magnitude of the vapor perturbation is only visible in the last 3 bins where average differences as high as –60 Wm$^{-2}$ can be reached. Similarly, there are substantial differences between adjacent dry and moist cells in cloud water path, drizzle water path and rain rate.


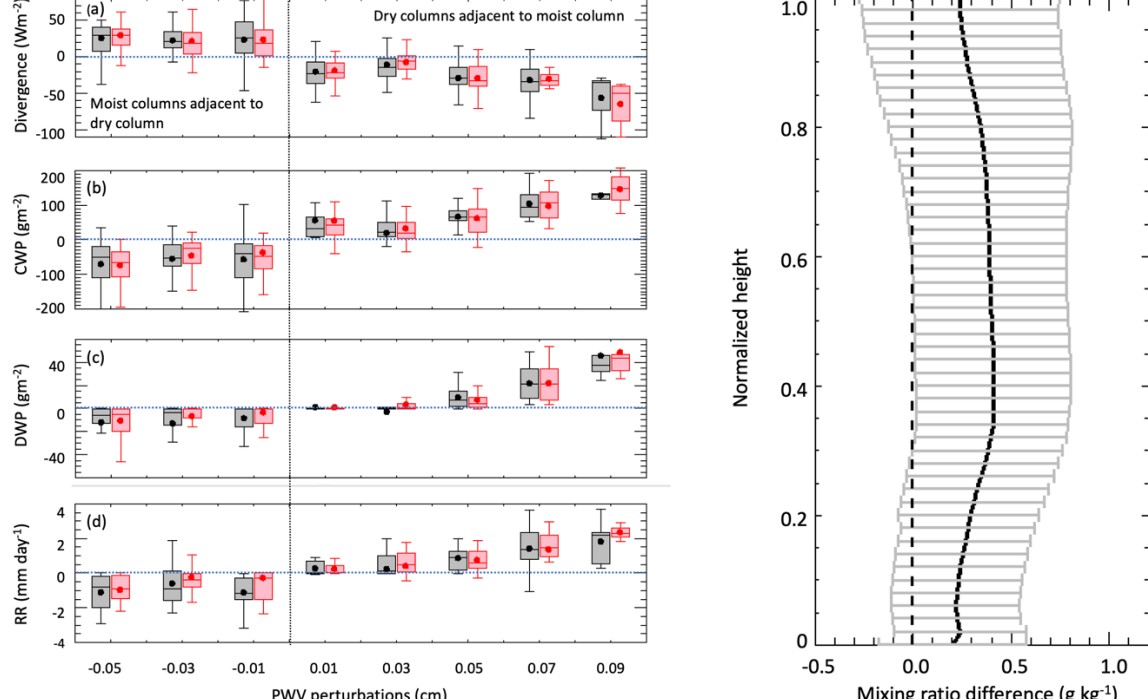

**Figure 11: Left:** ***Differences*** **in cloud top radiative cooling, cloud water path, drizzle water path, and rain rate at cloud base between each identified region (x-axis) and the 2 adjacent regions binned by the strength of the region's PWV perturbation, N=25, 22, 19, 15, 20, 15, 13, 5. Differences are computed as center-minus-preceding (black) and**
**center-minus-following (red) columns. Right: Vertical distribution of the mixing ratio** ***differences*** **between moist and dry columns.**

Moist cells always have higher liquid water path and precipitation. Small differences in drizzle water path and rain rate between positive and negative columns are still noticeable in the presence of weak vapor perturbations (< 0.03 cm). This
weak drizzle associated with a modest increase in the LWP may be driven by the stronger radiative cooling (~ 25 Wm$^{-2}$) in





the positive columns. In the presence of strong (> 0.03 cm) positive vapor perturbations, differences between moist and dry columns increase proportionally to the vapor perturbations indicating some correlation between the amount of moisture in the boundary layer and the amount of drizzle and rain rate in the mesoscale column. Mixing ratio differences between moist and dry columns are not uniformly distributed, instead are higher in the middle of the boundary layer as shown by the

average vertical distribution in Fig. 11.

## 5.2 Vertical air motion and moist static energy

Among the mechanisms proposed for mesoscale aggregation in shallow convection is one in which vertical air motion
promotes moisture transfer from the dry to the moist columns thereby increasing moisture variance. In the theoretical framework a mesoscale circulation is characterized by updrafts in the moist columns and subsiding dry air in the dry columns. With the help of the Doppler lidar we therefore examine the vertical velocities in moist and dry cells comparing it among adjacent regions in the sub-cloud layer. Only instances where at least 80% of the Doppler lidar readings are valid in a layer are used. This resulted in total updrafts (downdrafts) between $10^3$ and $10^4$ in the lower half of the boundary layer and
between $10^2$ and $10^3$ in the sub-cloud level. The samples were averaged inside each of the 143 moist and dry columns.
Figure 12 points to increased updraft frequency in the moist columns and increased downdraft frequency in the dry columns immediately below the cloud base.

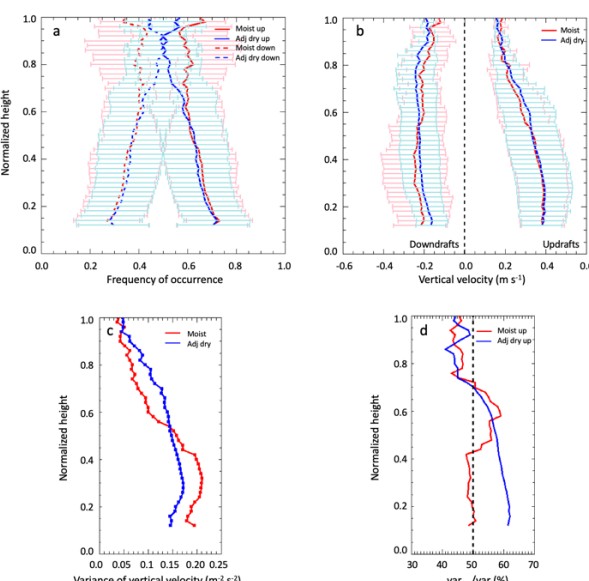

**Figure 12: (a) Frequency of updrafts and downdrafts, (b) vertical velocity, (c) variance of vertical velocity in moist**
**(red) and adjacent dry (blue) columns, (d) contribution of updrafts to total variance in moist and dry columns. The contribution of downdrafts can be estimated as 100-100\*var$_{up}$/var. The height is normalized to the cloud base height.**





In the same vertical range, downdrafts are slightly stronger in dry columns (10%-15%) but not in a statistically significant
way. Similarly, in the sub-cloud layer (Fig. 12 c) the variance of vertical velocity is 20%-60% higher in the dry columns than
in the moist columns. Looking at the relative contribution of updrafts and downdrafts in moist and dry columns, Fig. 12d
shows that updrafts in moist and dry columns contribute to the total variance in the bottom part of the boundary layer. In the
dry columns the contribution of downdrafts increases from 40% near the surface to 60% below the cloud base. Although the
shown differences are not statistically significant due to limited samples, these observations point towards the existence of a
subsidence region of increased turbulence in the sub-cloud layer of the dry patches where dry air may be entrained and
eventually mixed in the boundary layer, eventually reinforcing the drying. On the other end there may be a reinforcing of the
moistening in the moist columns through the lower troposphere and the lifting up of moist air in the sub-cloud layer.

To examine this part, we show in Fig. 13 curves of moist static energy (MSE) and virtual liquid static energy for the moist
columns and adjacent dry columns in the 2 days shown in Fig. 9. The MSE in moist and dry columns is similar between the
surface and the lower boundary layer. As previously mentioned in a well-mixed boundary layer conserved quantities such as
the liquid water potential temperature are constant, and the moist static energy is also conserved. Dry columns have similar
MSE compared to moist columns near the surface, but lower MSE in the upper part of the boundary layer, starting in the
layers immediately below the cloud base. This appears to confirm the concave nature of MSE and virtual liquid static energy
relationship shown by the previous modeling study of Bretherton and Blossey (2017).

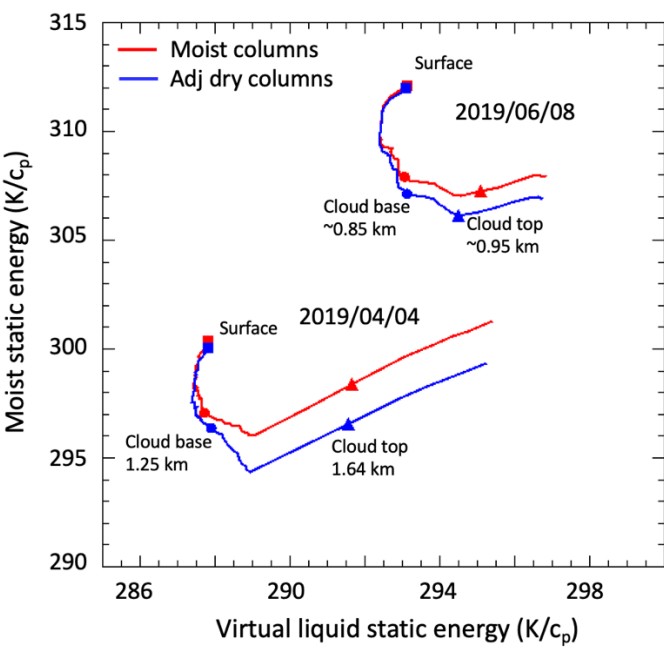

**Figure 13: Average moist static energy versus virtual liquid static energy in moist and adjacent dry columns of the 2
days shown in Fig. 9. Squares, circles, and triangle indicate surface, cloud base, and cloud top.**





**6 Discussion and conclusions**


In this work we have examined the factors that control boundary layer moisture at the ARM ENA site on a seasonal and daily temporal scale using 5 years of ground-based observations and reanalysis data. The boundary layer water vapor was highest during the summer months and lowest during the winter months. The mid-tropospheric humidity was also highest during the summer months and lowest during the winter months. The annual cycle of BL water vapor is anti-correlated to the

annual cycle of cloud and drizzle water path, highlighting the complex interaction between water vapor and clouds in the region. The lower mid-tropospheric humidity during the winter months, together with turbulence (Ghate et al. 2021) point towards turbulence being the primary controlling factor rather than water vapor in determining cloudiness in the region.

Monthly boundary layer water budgets were estimated using the mixed layer framework. The analysis shows that moistening from the latent hear flux is balanced mostly by large-scale advection of colder and dry air (~50%) followed by entrainment

drying (20%), and precipitation removal (~15%). Latent heat fluxes are enhanced in fall and winter resulting in average fluxes that are 26% stronger in winter and 22% weaker in summer compared to the annual mean. Although significant effort was spent in producing high-quality retrievals with low uncertainty bounds (Cadeddu et al., 2020), the moisture budgets could not be fully closed with an annual residual term of ~9 W m$^{-2}$, and larger monthly residuals. This primarily stems from the lack of PBL height and BL inversion strength observations at hourly timescales as they are derived from the radiosonde

launches made every 12 hours. Additional uncertainty is introduced due to direct measurement of BL entrainment rates that are derived from the mass-budget necessitating use of reanalysis predicted PBL depth. Measurements of surface and BL thermodynamic properties around the site such as those available at the ARM Southern Great Plains (SGP) site can possibly alleviate this issue.

On a daily temporal scale, we examined the mesoscale (10-100 km) organization of water vapor for 10 days characterized by

stratocumulus cloud conditions. Differences in water vapor mixing ratio, LWP, rain rate, cloud top radiative cooling, moist static energy, and vertical air motion between *adjacent* moist and dry mesoscale columns of vapor passing over the site were calculated. In these mostly drizzling systems, there are sharp differences between moist and dry patches with moist cells always displaying stronger cloud top cooling, higher LWP, and precipitation. Differences between moist and dry patches increase when the vapor perturbations are stronger suggesting some control of water vapor over the amount of precipitation

however, even in the presence of weak moist perturbations, there are detectable differences in precipitation and liquid water path suggesting a role of turbulence in drizzle initiation. Moist and dry patches present differences in vertical velocity with dry regions displaying more frequent downdrafts than moist regions immediately below the cloud base. In the same layers downdrafts in the dry columns appear 10%-15% stronger and the variance of vertical velocity 10%-28% higher. Finally, profiles of moist static energy in adjacent moist and dry columns show similar MSE in the low boundary layer decreasing in

the dry cells near the cloud base. Additionally, the departure of the MSE vs. liquid static energy from a straight line implies a

difference in the vertical mixing between moist and dry columns, conditions that would be favorable to the maintenance and amplification of the moisture variance and the mesoscale organization. These results suggest the presence of mesoscale convective aggregation in marine low clouds that is not represented in current ESM that have spatial resolution of 100 km or greater. However, as the ESM increase in spatial resolution (e.g., Caldwell et al. 2021), the cloud parameterizations will have

to account for the small-scale processes that cause this mesoscale aggregation.

Broadly the results presented herein are useful for future observational and modeling studies on low clouds conducted at the ARM ENA site. With the presence of mesoscale cellularity, and turbulence being higher in the winter as compared to the summer, we expect the aerosol effects on the low clouds to be more dominant in the summer months compared to the winter months. In addition, characterization of changes in cloud and rain properties due to aerosols will also need to involve proper

characterization of the water vapor fields as mesoscale changes in water vapor fields can potentially mask the aerosol effects.

## Data availability

The ground-based data used in this study were obtained from the Atmospheric Radiation Measurement (ARM) user facility, a U.S. Department of Energy (DOE) Office of Science user facility managed by the Office of Biological and Environmental

Research, and are available from https://www.arm.gov.

## Author contributions

MC prepared the manuscript with contributions from all authors. VG preprocessed, cleaned and calibrated the radar, ceilometer, and wind profiler data. MC performed the active and passive retrievals. DT provided the Raman lidar retrievals,

TS downloaded and preprocessed all datasets used in the analysis.

## Competing interests

The authors declare that they have no conflict of interest.

## Acknowledgments

Author Cadeddu is supported by the U.S. Department of Energy, Office of Science, Office of Biological and Environmental Research, Atmospheric Radiation Measurement Infrastructure, under Contract DE-AC02-06CH11357. Author Ghate was supported by the U.S. Department of Energy's (DOE) Atmospheric System Research (ASR), an Office of Science, Office of Biological and Environmental Research (BER) program, under Contract DE-AC02-06CH11357 awarded to Argonne

National Laboratory. We gratefully acknowledge the computing resources provided on Bebop, a high-performance computing cluster operated by the Laboratory Computing Resource Center (LCRC) at the Argonne National Laboratory.

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
