# Peer review of "Boundary layer moisture variability at the ARM Eastern North Atlantic Observatory during marine conditions"

_Atmospheric Chemistry and Physics, 2022_

## Author Comment (AC1)

Reviewer 1

**Summary**
This work examines surface-based observations at the Eastern North Atlantic (ENA) site on Graciosa Island in order to characterize the seasonal cycle and budget of boundary layer moisture. The manuscript is well-written and the authors have plenty of history and skill in working with these observations but I struggle to understand some of the important decisions that are made in the analysis process. My main concerns are related to the fundamental assumption of a well-mixed boundary layer at ENA and the severe limitations in selecting data to include only the extremes of a particular weather regime while also claiming to present an encompassing depiction of moisture variability at ENA. Both of these concerns are explained in my General Comments.

We thank the reviewer for the constructive comments that have led to changes and substantial improvements to the manuscript. Please find below our point-by-point responses to the comments. The comments are in black, and our responses to it are in green. Any changes to the article text are mentioned in blue.

**General Comments**
1. The authors assume a well-mixed boundary layer and justify their assumption by citing Albright et al. (2022) but the Albright paper was focused on the EUREC4A boundary layer, which is much closer to the Equator and actually located within the classical trade wind region. The ENA site often experiences decoupled boundary layer conditions and mid-latitude cyclone disturbances, invalidating this assumption that might be important for the conclusions presented in this work. The authors acknowledge that the assumption is potentially inaccurate but useful (lines 60-61). This manuscript should make more of an effort to justify this assumption. For example, this validation could take the form of showing that the decoupling index from ENA sondes is similar to EUREC4A or is generally low. Sonde profiles could also be shown, similar to Figure 2 in Albright et al. (2022). This could be included in the supplemental material, as the validation of the mixed-layer model for ENA is not really the focus of the manuscript.
Albright, A. L., S. Bony, B. Stevens, R. Vogel, 2022: Observed sub-cloud layer moisture and heat budgets in the trades. Journal of the Atmospheric Sciences, DOI 10.1175/JAS-D-21-0337.1

We thank both reviewers for bringing up this important point. The assumption of a well-mixed boundary layer is important to define the framework over which to study processes that affect clouds and moisture. The framework has been used for multiple decades now to understand the primary controls on the boundary layer vapor and (cloud) liquid field (e.g. Betts 1976 JAS for cumulus; and Brost et al., 1984 for stratocumulus). In our case, because we are using all data, there will be a mix of conditions, similar to the profiles shown by Lock et al. (2000 MWR). In theory the budget terms could be modified to include multiple mixed layers, or a mixed layer on top of a stable layer, or vice-versa. However, it is not possible to implement such a framework to the large data (~7 years) used in this study. It should be also noted that decoupling indices are usually calculated from radiosonde profiles, and at the ENA site we only have two radiosondes (00 and 12 UTC) per day, making it difficult to calculate budgets of a thermodynamically

decoupled boundary layer. This is primarily because any stable or mixed layers above the surface layers are transient and do not persist for more than 6 hours. However, the boundary layer inversion is omnipresent as it is primarily controlled by large scale subsidence. Lastly, in a decoupled boundary layer the surface moistening and the entrainment drying are mediated by boundary fluxes through another layer. Hence assuming a decoupled boundary layer is well-mixed will essentially cause an imbalance in the equation 1, thereby yielding a residual term. However, the seasonal and annual average residual term (Table 2) are far smaller than any of the calculated terms. This suggests that although the boundary layer may not be well-mixed, it can be assumed to be well-mixed at seasonal to annual timescales.

To address the reviewers concern in more detail we have therefore examined all radiosondes and radiosondes during marine conditions and calculated a Decoupling Index (DI) defined as:

$DI = (Z_{CB}-Z_{LCL})/Z_{CB}$

Profiles with DI < 0.25 are classified as strongly coupled and profiles with 0.25 < DI < 0.4 as weakly coupled.  With this definition more than half of marine profiles are strongly coupled and about 68% of the marine cases have DI < 0.4. Marine cases with low cloud base height (< 1.2 km) are statistically more coupled with 64% having DI < 0.25 and 80% having DI < 0.4.

|  | All | Marine | Marine with CB<1.2 km |
|---|---|---|---|
| Number of cloudy cases | 1825 | 681 | 469 |
| DI<0.25 | 739 (40%) | 351 (52%) | 301 (64%) |
| DI<0.4 | 1205 (66%) | 461 (68%) | 374 (80%) |

Selecting marine cases increases the percentage of coupled and weakly coupled cases compared to the entire dataset. The annual cycle of DI for the marine cases, shown in Fig. R1, shows higher variability in the summer months.

[Figure]

Fig. R1: Annual cycle of the decoupling index (DI) for (a) all marine cases and (b) marine cases with cloud base < 1.2 km.  The solid line indicates the 0.25 threshold (strongly coupled) and the dashed line indicates the 0.4 threshold.

To evaluate whether the inclusion of decoupled cases affects the moisture budget, we repeated the calculations of eq. 1 including only cases with cloud base height < 1.2 km. This is shown in Fig. R2. When compared with the new Fig. 7b, the main features of the results are not affected. However, the inclusion of decoupled profiles in the analysis (Fig. 7b) enhances the contribution of the entrainment fluxes to the total moisture sink from ~18% to ~25%.

[Figure]

Fig. R2: Seasonal fluxes calculated only for cases with cloud base height < 1.2 km where coupled conditions prevail. Colors are the same as in Figure 7b (Advection: Dark brown; Local tendency: Orange; Beige: Entrainment; Precipitation: Pink; Evaporation: Green)

| | DJF | MAM | JJA | SON | YEAR |
|---|---|---|---|---|---|
| $\dfrac{\mathcal{L}}{g}\hat{p}\dfrac{\partial\langle q_t\rangle}{\partial t}$ | -10.8± 68.5 (7.9%) | -7.9±62.1 (8.1%) | -17.3±59.6 (17.8%) | -18.7±68.8 (14.8%) | -14.9±64.6 (13.5%) |
| $\dfrac{\mathcal{L}}{g}\hat{p}\langle \boldsymbol{v}\cdot\boldsymbol{\nabla}q_t\rangle$ | -53.1±53.8 (39.0%) | -48.9±49.7 (50.0%) | -40.9±43.7 (42.2%) | -54.5±50.4 (43.2%) | -48.5±48.9 (43.8%) |
| $\dfrac{1}{g}\omega_e\Delta q_t$ | -23.6±12.6 (17.4%) | -17.6±8.9 (17.9%) | -15.1±9.0 (15.6%) | -16.2±9.1 (12.9%) | -17.8±9.8 (15.4%) |
| $\mathcal{L}\,P$ | -48.6±97.5 (35.7%) | -23.4±54.4 (23.9%) | -23.7±45.9 (24.4%) | -36.8±60.5 (29.2%) | -30.3±54.4 (27.3%) |
| E | 105.3±50.2 | 87.9±54.4 | 84.9±65.4 | 122.4±79.8 | 99.1±69.9 |
| Residual | -30.8 | -10.1 | -12.0 | -3.8 | -11.7 |

Table R1: **Seasonal average values and standard deviation of the budget components for cases with cloud base < 1.2 km. In parenthesis are the contributions of each negative term to the total boundary layer drying. Residuals are computed as the difference between source and sinks (Wm$^{-2}$).**

This point is now mentioned in the introduction Lines 65-68:
"The validity of the mixed layer framework has recently been shown to be sufficient to explain synoptic and monthly variability in the sub-cloud layer (Albright et al., 2022) however our dataset includes a mix of coupled and decoupled cases, and it is therefore important to understand how often the assumption of a well-mixed boundary layer is verified at the site, and how it affects the results."

Section 4 Lines 203-207
"To this end we examined 1825 soundings of which 681 where marine conditions and calculated a decoupling index (DI) defined as $(Z_{CB}-Z_{LCL})/Z_{CB}$. We then classified as strongly coupled cases with DI<0.25 and as weakly coupled cases with 0.25 < DI < 0.4. According to this classification most marine cases (68%) are weakly or strongly coupled. The decoupling index is generally smaller when the cloud base is lower and cases with cloud base < 1.2 km have DI<0.4 in 80% of the cases."

The uncertainty associated with this assumption is discussed in Section 4.3 at lines 327-331:
"At this point we are in the condition to evaluate the impact of including decoupled conditions in the analysis. We repeated the budget computations including only a subset with cloud base < 1.2 km which present mostly coupled conditions. The results showed a diminished contribution of the entrainment fluxes that decreased annually from 26% to 18%. It is therefore

likely that the inclusion of decoupled conditions in the analysis leads to an overestimation of the moisture sink due to entrainment fluxes."

In the discussion at lines 508-510:
"Although the majority of marine cases at the site can be classified as coupled or weakly coupled, inclusion of decoupled cases in the analysis introduces uncertainties leading to an overestimation of the contribution of entrainment fluxes to the budget."

2. The manuscript employs a strict definition of "marine conditions", discriminated solely on the basis of surface wind direction measured at the ENA site, which immediately discards 70% of the available observations. While the desire to eliminate effects of the island on observations is reasonable, given the goals of understanding the marine boundary layer, I wonder if the wind direction limitation also limits the analyses to certain weather regimes and biases the conclusions. Is the boundary layer either "not marine" or strongly affected by the land surface if the wind is from the west (wdir=270°)? There is likely some extra aerosol loading from the island's natural and human activity but does that significantly affect the moisture budget over such a short distance from shore? I would like to see some discussion of this potential issue either in Section 2.2 or Section 6 or both.

The reviewer is correct in the sense that the annual cycle of water vapor and liquid water path doesn't change much if we consider the entire dataset. However, inclusion of all cases makes a difference in the budget calculations as can be seen in the figure below. The surface fluxes are affected however the main difference is the contribution of the advection term that now includes not only the drying contribution from the North, but a moistening contribution from the South. Without separating the marine conditions, it becomes difficult to identify the large-scale mechanism that influences water vapor and clouds at the site. In addition, during southerly wind conditions, warm and moist air is advected over colder ocean water. Hence, by theory, this leads to negative surface fluxes and no cloud formation. Because our primary goal is the study of marine boundary layer clouds, the selection of marine conditions is essential to probe the related processes.

[Figure]

Fig. R3: Seasonal budget including all cases (marine and non-marine). Colors are the same as in Figure 7 (Advection: Dark brown; Local tendency: Orange; Beige: Entrainment; Precipitation: Pink; Evaporation: Green).

3. This manuscript seems to be more about moisture budgets during marine stratocumulus-topped boundary layers than simply "marine conditions". I think a change to the manuscript title is appropriate in order to accurately advertise the analyses according to the targeted weather/cloud type. The first paragraph of the discussion section also mentions, "we have examined the factors that control boundary layer moisture at the ARM ENA site on a seasonal and daily temporal scale using 5 years of ground-based observations and reanalysis data" but the really only the fully-overcast stratocumulus time periods with a particular surface wind direction were analyzed.

Probably due to poor phrasing in the initial version of the article, the reviewer got an impression that only overcast conditions were considered. The dataset used in the budget calculations includes all data (clear-sky, cloudy etc.), screened only for marine conditions (wind direction north of 90 and 310). The restriction on cloud fraction was imposed only for the small dataset used in the calculations of the adiabatic LWP that is now moved to the Appendix. We have clarified that in the text and also changed the title to better reflect the subset used. The adiabatic discussion is now moved to the Appendix following reviewer 2 suggestions. Thank you.

4. Many of the more interesting analyses and results are presented in the latter parts of the manuscript but there are so few samples due to inclusion of only 10 (hand-selected?) days. The conclusions formed from work explained in this manuscript would greatly benefit from an increased pool of data.

We entirely agree with the reviewer. The main constraint to the analysis was the effort necessary to run the AERIoe retrievals. We are now in the process of developing a retrieval from the Raman lidar that will provide a large dataset of high-resolution humidity profiles at 10 s time resolution. This new dataset will allow to expand and refine this analysis. Thank you.

**Specific Comments**

60: Are decoupled conditions common at ENA? See my General Comment 1 for more.

Assuming the radiosondes statistics as representative of the entire dataset it seems that coupled and weakly decoupled conditions constitute about 68% of the marine data. The effect of including decoupled conditions is now discussed.

144: This may well limit cases to marine environments but other environments exist at ENA so any conclusions are only for the marine state. Please be sure to make that clear throughout the paper. Only 30% of cases are "marine"?

Thank you, we have re-enforced this in the discussion and added to the title.

148: Why such a strict requirement? Is this cloud fraction computed from hourly data so there are 24 values per day? When the manuscript says, "In the following discussion only boundary layer clouds with cloud fraction from the ceilometer greater than 0.99 were selected…", does it mean all remaining analyses in the manuscript or just the brief remainder of Section 2 and Section 3?

We realize that this digression caused some confusion and we tried to clarify it in the revised version. For the adiabatic calculations we selected measurements coincident with radiosondes and for which the ceilometer reported cloudiness at least 99% of the time during the corresponding hour. This strict selection was imposed only to the adiabatic calculations, due to their sensitivity to cloud boundaries, and not to the budget analysis where all marine cases were considered. This discussion is now moved to the Appendix.

178: "a stronger contribution of the free troposphere to the total PWV in summer compared to winter". Examining Figure 3b, I find it hard to see whether the fractional contribution from z>3km to the total is higher in summer or winter. Certainly, the raw values of PWV(z>3km) are higher in the summer months. Maybe some clarification in 178 is appropriate to distinguish if you mean relative contribution or simply that the annual cycle of PWV(z>3km) peaks in the summer months, which was already stated in line 175.

We changed the phrasing now at Lines 172-175 to: "The annual cycle doesn't have the same amplitude in the upper and lower troposphere resulting in a stronger contribution of the free troposphere to the total PWV in summer compared to winter. The proportion of free tropospheric PWV to the total amount ranges from 14% in February to 20% in June."

We also add here in Fig. R4 for the reviewer a plot of the fraction of upper tropospheric water vapor (> 3 km) to the total water vapor. The upper troposphere contributions vary from ~16% in winter to ~23% in summer. The values are: 0.169722      0.154620      0.165167      0.178396 0.203754    0.221886    0.247377 0.203362    0.188243    0.172586    0.161159    0.178956.

[Figure]

Fig. R4: Fraction of PWV > 3 km to total PWV.

Figure 3: The annual cycle in PWV would be easier to see in Figure 3a if the y-axis limit was reduced to 5 cm. Also, why are the lower y-limits not 0 in all cases?
Figure 3 (now figure 2) was changed as requested. We slightly extended the y axis to show the lower end of the error bar. There are no negative values.

Figure 3: It would be nice to show how many observations are used in each month as the top panel of this figure. Annual cycles in wind direction and cloud type could play a role in the interpretation of these results.
We agree with the reviewer and report now the number of samples in Fig. 3 (now Figure 2). On the left axis is the number of radiosondes samples: 73 79 79 106 71 135 152 148 132 173 143 51

and on the right axis the number of retrievals: 647 578 775 994 536 1056 1300 1338 1283 1793 1372 384. The reviewer is correct that there is a seasonal dependence of the samples probably due to wind direction.

207: This actually appears to be a very *small* dataset of only 304 points! At this point in the manuscript, no ERA5 data are yet used, right? So why have the analyses been averaged up to hourly resolution? This is likely an overly-strict limitation that throws away too much valid data. As I understand it, Figure 4 is showing only 304 out of an initially-available 52608 points, only about one half of one percent of the total data. I realize that radiosondes aren't released every hour, but the cloud fraction>0.99, wind direction, and weak precipitation requirements are likely overly strict and therefore likely to bias the conclusions of these analyses.

At the beginning we were also surprised at the exiguity of the adiabatic sample. Looking at the numbers however, the reasons for this dramatic selection appear more reasonable: The adiabatic calculations can be done only where we have cloud top and cloud base well defined. Because there are only 2 launches per day, we have about 2000 marine cases. Out of these the cloud top height was found in 1190 marine cases. When we combine the ceilometer including the cloud fraction restriction, MWR, and sondes, we find that all data were present and valid in about 300 cases. Probably the only way to expand the dataset for the computation of the adiabaticity is to use the KAZR to identify the cloud top height and have continuous retrievals of thermodynamic properties This would likely increase the dataset but not of much.

219-220: Many of the cases, even for relatively thick (>500 m) clouds, have adiabaticity considerably greater than one, with some as high as 150% adiabatic. Please provide more discussion and validation of the cloud boundary argument. If these cases simply appear to be superadiabatic due to uncertainties in cloud boundaries, that uncertainty will likely also affect the cases that appear sub-adiabatic. Can these "superadiabatic" cases result from the microwave radiometer seeing elevated large rain drops instead of only smaller cloud drops? More discussion is needed here because this occurs for a large portion of the limited dataset even for relatively thick clouds.

We thank the reviewer for this comment. As the reviewer points out, the largest uncertainties are due to liquid water path and cloud boundaries. When we examined the calculations, we also realized that we had used the total liquid water path instead of the cloud liquid water path. When we changed this, we obtained 55 superadiabatic cases or ~14% instead of the previous 19%.

After this change, we also examined the effect cloud boundaries. Increasing the cloud thickness of 50 m decreased the mean $f_{ad}$ from 0.56 to 0.45 or 20% of the calculated value. Similarly, decreasing the cloud thickness of 50 m increased the mean $f_{ad}$ from 0.56 to 0.67 or 50%. It is therefore likely that, in addition to the uncertainty in LWP, uncertainty in cloud boundaries affect the calculations. We replaced the figure (now Fig. A2) to reflect the changes and added the following discussion in Appendix A lines 619-622 where the adiabaticity is now discussed:

"Increasing the cloud thickness of 50 m decreased the mean $f_{ad}$ from 0.56 to 0.45 or 20% of the calculated value. Similarly decreasing the cloud thickness of 50 m increased the mean $f_{ad}$ from 0.56 to 0.67 or 50%. It is therefore likely that, in addition to the uncertainty in LWP, uncertainty in cloud boundaries affect the calculations."

230: Does the liquid potion (ql) also include rain water mixing ratio? I assume it does from equation 1 but it would be helpful to be explicit about it in line 230.

Yes, we include rainwater mixing-ratio. Added "including rain" at line 210.

254: The MBL height is likely not constant during the 12 hours containing a given PBL measurement. Would it be better to use a moving polynomial interpolation?

It could be done but considering that at the end we average again it may not have an effect on the overall results. If we were looking at daily time scales it may make a difference though.

Line 273: If you've limited the analyses to when wind direction indicates "marine conditions", why are there *any* cases with v>0? Disagreement between ENA observations and ERA5?

Marine conditions were determined using *hourly averaged* MET measurements at the site. In each individual met hours there could be wind in other directions. In fig 5 (now Fig. 3) we plot hourly ECMWF data. Individual hours from the ECMWF dataset could also have occasional prevailing wind in other directions as shown by the whiskers extending to positive values, the mean values are consistent with the average MET measurements at the site. The ECMWF data are averages over a 25 km x 25 km grid and mostly signify the background wind conditions. Hence there are small number of cases where the local wind as measured by the surface met station is from the North, while the background prevailing wind is from the South. Thank you.

276: Why omit the extremes? What evidence do you have that these extremes are simply instrument noise instead of real events that should be included in the analyses?

We agree that it is hard to distinguish between real rain extremes and effects due to splashing of raindrops at the surface. It is also likely that during hours of extreme precipitation other sensors, such as the MWR have increased uncertainty. Nonetheless we repeated the budget calculations with all precipitation data and have updated figure 7 (now Fig. 5) and table 2. The conclusions are unchanged, but there is an increase in the precipitation contribution during winter months. Uncertainties in the rain rate measurements are one of uncertainty components of the budget.

| | DJF | MAM | JJA | SON | YEAR |
|---|---|---|---|---|---|
| $\dfrac{\mathcal{L}}{g}\hat{p}\dfrac{\partial\langle q_t\rangle}{\partial t}$ | -8.4± 67.5 (6.2%) | -7.6±62.2 (7.1%) | -13.2±58.5 (13.7%) | -9.6±66.5 (7.5%) | -10.1±63.5 (8.9%) |
| $\dfrac{\mathcal{L}}{g}\hat{p}\langle v \cdot \nabla q_t\rangle$ | -54.1±51.3 (40.0%) | -51.9±47.5 (48.9%) | -40.1±41.6 (41.6%) | -56.9±51.2 (44.3%) | -50.4±48.2 (44.2%) |
| $\dfrac{1}{g}\omega_e\Delta q_t$ | -27.8±14.1 (20.6%) | -25.5±12.1 (24.0%) | -22.4±12.5 (23.3%) | -31.5±15.9 (24.5%) | -25.8±14.3 (23.4%) |
| $\mathcal{L}\,P$ | -44.9±89.9 (33.2%) | -21.1±46.9 (19.9%) | -20.7±60.5 (21.4%) | -30.5±63.5 (23.7%) | -26.8±61.8 (23.6%) |
| E | 119.7±57.7 | 106.7±62.2 | 87.1±48.1 | 141.2±68.5 | 112.7±63.3 |
| Residual | -15.4 | 0.6 | -9.4 | 12.6 | -1.4 |

**Table 2: Seasonal average values and standard deviation of the budget components. In parenthesis are the contributions of each negative term to the total boundary layer drying. Residuals are computed as the difference between source and sinks (Wm$^{-2}$).**

301: Air density is not constant in the boundary layer. Is there confirmation that the vertical change in atmospheric density matters much less than changes in PBL height? Or do you mean that the profile of air density is relatively constant in time?

We mean that the air-density is horizontally constant but changes vertically. This assumption is needed for deriving mixed layer budget equation, and is regularly applied while utilizing mixed layer budgets (e.g. Kalmus et al., 2011 J. Climate; Ghate et al., 2019 QJRMS). Thank you.

387: How were these cases chosen? The manuscript says "The selected days displayed persistent boundary layer cloudiness and at times precipitation" but how were they identified? By eye? Some thresholds on cloudiness?

The cases were a subset of a larger sample of marine stratocumulus cases. There were not additional constraints, besides data availability. The exiguity of the dataset is mostly due to the complexity of running all retrievals, especially the humidity retrievals that require AERI, Raman lidar, and MWR. We are working on a larger dataset for which we are developing a Raman lidar retrieval of water vapor at 10 s time resolution.

397: Why are daily averages used? Do the winds not change at all during each of the 10 days? These cloud base winds come from ERA5, right?

The wind speed at the cloud base was derived from the interpolated sondes and the variability of the wind speed during the selected days was low. We now add this information at line 383. Below is a table with the average wind speed at cloud base and standard deviation for the selected days.

| Date | Wind speed at cloud base, m s$^{-1}$ |
|---|---|
| 20170316 | 9.0±0.6 |
| 20170317 | 8.2±1.0 |
| 20180805 | 4.5±0.5 |
| 20180904 | 5.5±1.2 |
| 20180331 | 5.9±1.8 |
| 20190404 | 11.0±1.5 |
| 20190604 | 7.8±1.6 |
| 20190608 | 5.1±2.7 |
| 20190625 | 6.3± 0.9 |
| 20190626 | 8.3±0.7 |

400: So there are 345 mesoscale chunks during the 10 selected days? The chunks from within a given day have the same amount of temporal averaging but there are different temporal averaging times between the 10 days?
The chunks have the same amount of spatial (10 km) averaging, but, because of the different wind speed, the temporal averaging times vary between the 10 days.

433: In line 428, the manuscript states that "moist and dry neighboring columns … were compared to those of its preceding and following neighbors. For clarification, are all of the 345 mesoscale perturbations considered but only 143 columns had durations longer than 10 minutes or was there a requirement for Figure 10 that only moist columns that were neighbored by dry columns be considered?
All perturbations were considered, and they all alternated positive and negative columns.

Figure 11: There are really few cases here. I think the filtering is too strict. How can we know that these 5 cases constitute a representative sample the full population for PWV perturbations from 0.8 to 1.0 cm?
We agree that the largest perturbations are not representative because of the scarcity of samples and reinforce this concept in the text. We would like to keep this part that we feel is important and are working on a larger dataset. The extended dataset requires the development of new high resolution vapor retrievals from the Rama lidar. We added the following statement at line 430: "The few cases in the last bin are not sufficient to provide a valid sample from which to draw definite conclusions."

469: Please remind the reader what papers have proposed and promoted this mechanism.
Added Bretherton, C. S. and P.N., Blossey, 2017 at line 457.

473: Why might the doppler lidar retrievals be invalid in a given layer? Could these observational limitations imposed by the doppler lidar result in a biased view of the vertical motion?

The Doppler Lidar signal attenuates vertically near cloud base, which usually is fluctuating. Hence there are instances where samples will be available near the surface and most of the sub-cloud layer, while very few samples near the cloud base. Hence the averages calculated from these samples can potentially lead to incorrect conclusions. To get around this issue, we have only averaged heights where at least 80% of the time Doppler Lidar samples were available.
The sentence is changed to the following, "Only heights where at least 80% of the Doppler lidar readings are valid were used in calculating these averages." Now line 460-461.

Figure 12: It would be simple and helpful to add uncertainties to (c) and (d) by subsampling/resampling techniques, which would strengthen the claims about differences between the two distributions.
We reran the 2 figures for 3 subsets of approximately half the original size and now report in the figure the boundaries as error bars.

538: Other papers have suggested this, too, right? They should be cited here.
We have cited almost all the articles showing convective aggregation earlies in the section. We therefore added: "as hypothesized in the previously cited works." At line 535-536. Thank you.

**Technical Corrections**
238: It would be best if you used either "3$^{rd}$ and 4$^{th}$" or "third and fourth". -Done

418: Both the LWP and precipitation increase when columns are moist so you do not need the "respectively". Changed-Thank you

418: Parentheses are for clarification and references, not the exact opposite of what was just stated. You could remove the content of the parentheses and the sentence would be much clearer to the reader. Sentences like, "Moister columns correspond to regions of increased liquid water path and precipitation." and "Increased moisture is associated with increased liquid water path and precipitation" are understood more easily. Additionally, in the preceding paragraph and in the following sentence, parentheses are used for clarification instead of opposition. - Changed

436: A dryer is a household appliance for making wet close dry. You want "drier". I confuse these two often, too. Done, thank you for catching this.

474: Why is "downdrafts" in parentheses? Removed, it was probably a residue from a previous version.

---

## Author Comment (AC2)

Reviewer 2.

This paper analyses the variability of boundary layer moisture at the ARM ENA site using ground-based observations and ERA5 reanalysis data. The authors compute mixed-layer water budgets at monthly timescales and analyze the respective contributions of the different terms, and then assess mesoscale moisture variability on sub-daily timescales and their relationships to updrafts, liquid water path and precipitation. The analyses seem sound, and the paper is well written and quite easy to follow. My major concerns are that (i) many assumptions are not justified or discussed, (ii) I did not understand if the authors just focus on stratocumulus or also shallow cumulus conditions, (iii) the story and sequence of analyses is hard to anticipate, and the novelty and connection among different sections is not really clear, and (iv) uncertainties are not systematically quantified.

These comments, along with some more minor comments, are addressed in more detail below.

We thank the reviewer for the constructive comments that have led to changes and substantial improvements to the manuscript. Please find below our point-by-point responses to the comments. The comments are in black, and our responses to it are in green. Any changes to the article text are mentioned in blue.

**Major comments:**

1.**Justification & discussion of assumptions:** For many of the assumptions made here, I miss a justification or discussion. For example, are the boundary layers analyzed really well-mixed up to the boundary layer top? For stratocumulus conditions I'd assume this is the case, but for decoupled shallow cumulus conditions, only the sub-cloud layer is well mixed. (See also comment 2 regarding the cloud regimes below). Could the authors demonstrate the well-mixedness of their boundary layers, and discuss how different cloud regimes might affect the budgets?

We thank both reviewers for pointing this out. We now discuss this topic and infer the uncertainty due to the inclusion of cases that are not well mixed. We repeat here the discussion provided for the 1st reviewer.

The assumption of a well-mixed boundary layer is important to define the framework over which to study processes that affect clouds and moisture. The framework has been used for multiple decades now to understand the primary controls on the boundary layer vapor and (cloud) liquid field (e.g. Betts 1976 JAS for cumulus; and Brost et al., 1984 for stratocumulus). In our case, because we are using all data, there will be a mix of conditions, similar to the profiles shown by Lock et al. (2000 MWR). In theory the budget terms could be modified to include multiple mixed layers, or a mixed layer on top of a stable layer, or vice-versa. However, it is not possible to implement such a framework to the large data (~7 years) used in this study. It should be also noted that decoupling indices are usually calculated from radiosonde profiles, and at the ENA site we only have two radiosondes (00 and 12 UTC) per day, making it difficult to calculate budgets

of a thermodynamically decoupled boundary layer. This is primarily because any stable or mixed layers above the surface layers are transient and do not persist for more than 6 hours. However, the boundary layer inversion is omnipresent as it is primarily controlled by large scale subsidence. Lastly, in a decoupled boundary layer the surface moistening and the entrainment drying are mediated by boundary fluxes through another layer. Hence assuming a decoupled boundary layer is well-mixed will essentially cause an imbalance in the equation 1, thereby yielding a residual term. However, the seasonal and annual average residual term (Table 2) are far smaller than any of the calculated terms. This suggests that although the boundary layer may not be well-mixed, it can be assumed to be well-mixed at seasonal to annual timescales.

To address the reviewers concern in more detail we have therefore examined all radiosondes and radiosondes during marine conditions and calculated a Decoupling Index (DI) defined as:

$DI = (Z_{CB} - Z_{LCL})/Z_{CB}$

Profiles with DI < 0.25 are classified as strongly coupled and profiles with 0.25 < DI < 0.4 as weakly coupled. With this definition more than half of marine profiles are strongly coupled and about 68% of the marine cases have DI < 0.4. Marine cases with low cloud base height (< 1.2 km) are statistically more coupled with 64% having DI < 0.25 and 80% having DI < 0.4.

|  | All | Marine | Marine with CB<1.2 km |
|---|---|---|---|
| Number of cloudy cases | 1825 | 681 | 469 |
| DI<0.25 | 739 (40%) | 351 (52%) | 301 (64%) |
| DI<0.4 | 1205 (66%) | 461 (68%) | 374 (80%) |

Selecting marine cases increases the percentage of coupled and weakly coupled cases compared to the entire dataset. The annual cycle of DI for the marine cases, shown in Fig. R1, shows higher variability in the summer months.

[Figure]

Fig. R1: Annual cycle of the decoupling index (DI) for (a) all marine cases and (b) marine cases with cloud base < 1.2 km. The solid line indicates the 0.25 threshold (strongly coupled) and the dashed line indicates the 0.4 threshold.

To evaluate whether the inclusion of decoupled cases affects the moisture budget, we repeated the calculations of eq. 1 including only cases with cloud base height < 1.2 km. This is shown in Fig. R2. When compared with the new Fig. 7b, the main features of the results are not affected. However, the inclusion of decoupled profiles in the analysis (Fig. 7b) enhances the contribution of the entrainment fluxes to the total moisture sink from ~18% to ~25%.

[Figure]

Fig. R2: Seasonal fluxes calculated only for cases with cloud base height < 1.2 km where coupled conditions prevail. Colors are the same as in Figure 7b (Advection: Dark brown; Local tendency: Orange; Beige: Entrainment; Precipitation: Pink; Evaporation: Green)

|  | DJF | MAM | JJA | SON | YEAR |
|---|---|---|---|---|---|
| $\dfrac{\mathcal{L}}{g}\hat{p}\dfrac{\partial\langle q_t\rangle}{\partial t}$ | -10.8± 68.5 (7.9%) | -7.9±62.1 (8.1%) | -17.3±59.6 (17.8%) | -18.7±68.8 (14.8%) | -14.9±64.6 (13.5%) |
| $\dfrac{\mathcal{L}}{g}\hat{p}\langle \boldsymbol{v}\cdot\boldsymbol{\nabla}q_t\rangle$ | -53.1±53.8 (39.0%) | -48.9±49.7 (50.0%) | -40.9±43.7 (42.2%) | -54.5±50.4 (43.2%) | -48.5±48.9 (43.8%) |
| $\dfrac{1}{g}\omega_e\Delta q_t$ | -23.6±12.6 (17.4%) | -17.6±8.9 (17.9%) | -15.1±9.0 (15.6%) | -16.2±9.1 (12.9%) | -17.8±9.8 (15.4%) |
| $\mathcal{L}\,P$ | -48.6±97.5 (35.7%) | -23.4±54.4 (23.9%) | -23.7±45.9 (24.4%) | -36.8±60.5 (29.2%) | -30.3±54.4 (27.3%) |
| E | 105.3±50.2 | 87.9±54.4 | 84.9±65.4 | 122.4±79.8 | 99.1±69.9 |
| Residual | -30.8 | -10.1 | -12.0 | -3.8 | -11.7 |

Table R1: **Seasonal average values and standard deviation of the budget components for cases with cloud base < 1.2 km. In parenthesis are the contributions of each negative term to the total boundary layer drying. Residuals are computed as the difference between source and sinks (Wm$^{-2}$).**

This point is now mentioned in the introduction Lines 65-68:
"The validity of the mixed layer framework has recently been shown to be sufficient to explain synoptic and monthly variability in the sub-cloud layer (Albright et al., 2022) however our dataset includes a mix of coupled and decoupled cases, and it is therefore important to understand how often the assumption of a well-mixed boundary layer is verified at the site, and how it affects the results."

Section 4 Lines 203-207
"To this end we examined 1825 soundings of which 681 where marine conditions and calculated a decoupling index (DI) defined as $(Z_{CB}-Z_{LCL})/Z_{CB}$. We then classified as strongly coupled cases with DI<0.25 and as weakly coupled cases with 0.25 < DI < 0.4. According to this classification most marine cases (68%) are weakly or strongly coupled. The decoupling index is generally smaller when the cloud base is lower and cases with cloud base < 1.2 km have DI<0.4 in 80% of the cases."

And the uncertainty associated with this assumption is discussed in Section 4.3 at lines 327-331:
"At this point we are in the condition to evaluate the impact of including decoupled conditions in the analysis. We repeated the budget computations including only a subset with cloud base < 1.2 km which present mostly coupled conditions. The results showed a diminished contribution of the entrainment fluxes that decreased annually from 26% to 18%. It is therefore

likely that the inclusion of decoupled conditions in the analysis leads to an overestimation of the moisture sink due to entrainment fluxes."

In the discussion at lines 508-510:
"Although the majority of marine cases at the site can be classified as coupled or weakly coupled, inclusion of decoupled cases in the analysis introduces uncertainties leading to an overestimation of the contribution of entrainment fluxes to the budget."

I understand that ERA5 data is necessary to complement the observations, but did the authors check whether the moisture profiles are consistent with the radiosonde and Raman lidar profiles? E.g., are the moisture profiles consistent enough that ERA5 can be used to computed the gradient? Even if this can be checked only for a limited data sample, it would greatly increase confidence in the approach. Similarly, for the local tendency and the PBL heights, could the Raman lidar profiles be used to check the hourly variability and a potential diurnal cycle in the terms that would be missed with the twice daily radiosondes?

ECMWF moisture profiles were used for the calculations of the large-scale advection of qv and for the advection of the PBL height used in the mass budget equation. For everything else (i.e., PBL height for the integration in eq. 1, qv for the local tendency, and $\Delta q_t$) we used radiosondes. We agree that it would be great to have Raman lidar profiles. Unfortunately, we don't have those yet, but are developing the retrievals to derive RL profiles at 10 s resolution. We provide in Fig. R5 a comparison between ECMWF and radiosondes expressed as mean and standard deviation of differences and the mean difference expressed as a percentage of the average mixing ratio at the site. Statistically the ECMWF seem to underestimate the mixing ratio of about 10-15%. The differences are more pronounced near the top of the boundary layer where the humidity gradient is often located.

We added this discussion in the paper in section 2.1, Line 115-119:
"ECMWF profiles were compared to the local soundings between 2015 and 2020 and found to underestimate the mixing ratio of about 10% with standard deviation of 1.-1.6 gkg⁻¹ between 0 and 3 km with a maximum underestimation of 15% at the PBL height. The ECMWF profiles are therefore suitable for the estimation of the vapor advection component of the budget but not suitable for the estimation of the PBL height. For this purpose, we use radiosondes."

[Figure]

[Figure]

Fig. R5: (a) Mean difference between RS and ECMWF (black) and standard deviation around the mean difference (red) for N=4139 soundings between 2015 and 2020. (b) Mean difference between RS and ECMWF expressed as a percentage of the average mixing ratio.

**2. Cloud regimes:** As alluded to in Sec. 1 and 2, both stratocumulus and shallow cumulus conditions are frequent at ENA. But these cloud regimes are associated with coupled vs. uncoupled boundary layers, which is a relevant difference for this study. Except for Sec. 5, which clearly addresses stratocumulus conditions, only in L147 a cloud criterion is mention: "In the following discussion only boundary layer clouds with cloud fraction from the ceilometer greater than 0.99 were selected (total of 3580 hours)." So does this mean that the entire discussion of the budgets focuses on stratocumulus conditions with ~100% cloud cover? And does the 0.99 threshold refers to hourly values? Please clarify, and discuss more prominently.

We realize that this inclusion caused confusion and was mentioned by the other reviewer as well. The strict condition of overcast cloudiness was set only for the purpose of adiabatic computations for the reason that the adiabatic computations are very sensitive to the cloud boundaries. We understand that the digression made the paper hard to follow and have moved it to the appendix in the revised version. This way it is clear that the restrictive conditions only apply to the data in the Appendix.

**3. Story, structure of the paper and novelty:** I found it hard to anticipate the story and the structure of the paper from the abstract and the introduction. For example, Sec. 3.1 on the cloud adiabaticity seems rather peripheral and came quite unexpectedly. And at first, I expected just an analysis of the water vapor budget, and then I realize the paper focuses on the total water budget, and also constrains the mass budget. So it would have helped me if the story of the paper was clearer and if the reader's expectation was a bit more guided.

Following the reviewer's comment, we have moved the discussion of cloud adiabaticity and retrieval discussion to the Appendix. We now also clarify better at the beginning that we are considering the total water budgets at Line 80-81 (vapor and liquid, including rain) and we use all marine cases without cloud selection.

In a similar spirit, I missed the coherence and connections between the different Sections. How do the monthly mixed-layer budget analyses connect to the analyses of the mesoscale variability? Could you construct mixed-layer budgets also on shorter timescales to connect to the monthly budgets and understand how the terms contribute differently at different timescales?

Thank you for raising this issue. The purpose of the analysis was to understand the role of moisture at the site at different temporal and hence spatial scales. Clouds and liquid water path exhibit a distinct annual and diurnal cycle as reported by previous studies (e.g. Ghate et al., 2021 JAMC). However, boundary layer water vapor only exhibits a distinct annual cycle, not a diurnal cycle. In addition to the annual cycle, water vapor also exhibits a mesoscale variability. Both the annual and mesoscale variability influence clouds and precipitation, the annual variability being driven by large scale advection and the mesoscale variability being driven by local processes. Both variabilities are hence explored in this article.

We added these considerations in the introduction Lines 73-76: "Water vapor, unlike LWP, does not exhibit a diurnal cycle, therefore the annual and mesoscale variability are the primary modes through which the interaction between boundary layer vapor and cloud processes can be examined. Both modes influence clouds and precipitation, the annual variability being driven by large scale advection, as shown later, and the mesoscale variability being driven by local processes."

About the second question, calculating the budgets on diurnal timescales would have required PBL height at hourly timescales that we lack due to the sparsity of radiosonde launches. However, we would like to note that, if suitable data were available, a study on diurnal budgets in different seasons would illuminate the role played by different processes in determining LWP, because LWP, rain rate and entrainment exhibit a distinct diurnal cycle.

I would also recommend the authors to highlight more explicitly what is new in the paper. How does the novel retrieval used, which better separates cloud and drizzle contributions to TLWP, affect the robustness of your analyses and conclusions? What are the novel insights gained with the ENA data here compared to previous ML budgets? In L541 it is mentioned that 'the results presented herein are useful for future observational and modeling studies on low clouds conducted at the ARM ENA site', but can you be a bit more explicit?

We thank the reviewer for this observation that gave us the opportunity to better discuss this aspect. The derived moisture budgets utilized high-quality retrievals that have low uncertainty thereby giving us confidence that the results represent physical processes rather than instrumentation issues. As shown in Table 1 the analysis utilized many instruments each

requiring careful calibration, denoising, cleaning, and (some) retrieval development. The consistency of the results among the instruments and the physically meaningful scenarios emerging from the mixed layer budget analysis highlight the quality and low uncertainty of the data, and the feasibility of the procedure used to calculate the budgets. Among the insights allowed by the new retrievals are the ubiquitous presence of drizzle throughout the year even in seasons when the average LWP is low, and the relatively more pronounced drizzle annual cycle compared to the LWP cycle, pointing to the fact that LWP is only one of the factors influencing drizzle formation. The fact the drizzle LWP has an annual cycle that is distinct from the cloud LWP points also to the importance of looking at processes seasonally. We think these results will be very useful in guiding our next study when we will analyze the combined effects of aerosols and turbulence in the development of precipitation, and we hope they will be useful to others as well. To our knowledge this aspect hasn't been fully examined yet.

To our knowledge, the present analysis is also the first study to characterize and determine the controls of moisture variability at a *subtropical* marine location such as the ENA site. Previous studies performing moisture budgets either utilized reanalysis data (e.g Wood and Bretherton, 2006), or coarse satellite data (e.g., Kalmus et al., 2012), or were of limited duration (e.g Caldwell et al. 2005; Albright et al. 2022; Brost et al. 1982). Hence it is difficult to make direct comparison to these studies. However, our derived entrainment rates, and precipitation fluxes are in close agreement to those reported by Caldwell et al., 2005 and Kalmus et al., 2012.

Finally with regard to the last question, the current analysis will be useful for future modeling and observational studies in various ways. For example, future studies aimed at retrieving or simulating entrainment rates should expect higher values in the winter months compared to summer months. Studies aimed at simulating clouds at the ENA site using traditional Lagrangian LES framework should utilize thermodynamic profiles of parcels before it gets advected over the site as advection contributes ~50% to the total moisture budget at the site. Finally similar moisture budget analysis performed using output from an Earth System Model (ESM) will inform whether the ESM is accurately simulating water vapor in the region and its sources and sinks at different spatial and temporal scales.

We have added some of these considerations in the conclusions:

Lines 496-500 Unlike LWP, water vapor at the site doesn't present a diurnal cycle but presents an annual and mesoscale variability that is strongly connected with cloud and precipitation at the site. To our knowledge, the present analysis is also the first study to characterize and determine the controls of moisture variability at a subtropical marine location, such as the ENA site, using a long-term dataset of ground-based data.

Lines 539-544 "As a final consideration we spend a few words to highlight how the separation of cloud and drizzle water path in the new retrievals reveals the ubiquitous presence of drizzle throughout the year even in seasons when the average LWP is low. By looking only at the total LWP, only a weak annual variation appears, however the drizzle LWP shows a more pronounced seasonal variability, pointing to the fact that LWP is only one of the factors influencing drizzle

formation. These results can be useful for future observational studies aimed at understanding the combined effects of aerosols and turbulence in the development of precipitation, pointing to the importance of looking at processes seasonally."

Lines 549-551: "Finally, the annual regional moisture budget resulting from an Earth System Model (ESM) will inform whether the ESM is accurately capturing the water vapor variability in the region and its sources and sinks at different spatial and temporal scales."

**4. Uncertainty quantification:** I missed a quantification of the uncertainty of the different terms in equations (1) and (3). Also, can you briefly say in L133 how the uncertainties of the retrievals are estimated?
We have added an explanation of the uncertainty of each term in section 4.1.

Lines 236-238: "Uncertainty in the estimation of PBL height from radiosondes is estimated to be 100-200 m (Sivaraman et al., 2013) this is a lower limit estimate in our case because the PBL height is kept constant for 12 hours."

Lines 247-249: "The uncertainty of this term is driven by the uncertainty of the microwave radiometer retrievals that is estimated 0.5 mm for water vapor and 15 $gm^{-2}$ for liquid water path. This translates in an uncertainty of less than 0.25 g/kg for the average $q_t$."

Lines 259-261: "The uncertainty of the advection term is hard to estimate. From a comparison of ECMWF and radiosondes profiles at the ENA the mixing ratio uncertainty is expected to be around 15% with higher uncertainty near the top of the boundary layer, where the humidity gradient is located."

The SPARCL precipitable water vapor and liquid water path uncertainties are derived from the a posteriori covariance of the optimal estimation.
We added at line 140: "Uncertainties in the water vapor and liquid water path retrievals are estimated from the a posteriori covariance information obtained with the optimal estimation retrieval…"

**Minor comments:**

- Retrievals section 2.3: I think this section could be written a bit more concisely. From one paragraph to the other, it seems to jump from one algorithm (with unfamiliar acronym for me) to another. Also, I think it could be worthwhile to present the comparison of SPARCL and MWRRET (L141 onwards) in an appendix, not to depart from the main story of the paper too much.

  We thank the reviewer for this suggestion and moved the comparison to the Appendix as suggested.

- Use of commas: I missed a lot of commas throughout the manuscript, e.g.:
    - L113: were derived, cases that ....
    - L134: For ten selected cases of weakly precipitating marine stratocumulus clouds, vertical profiles of water
    - L143: Traditionally, the total liquid water path retrieved by radiometers is assumed to represent the cloud water path. However, in the presence
    - ... Please check the entire manuscript carefully.

    We checked the manuscript- Thank you

- References: I'm not sure which program (if any) the authors use for the references, but it's not consistent and sometimes erroneous. E.g. Zhou and Bretherton 2019 is cited differently in L381 and L432. Also, the reference 'Shultz and Stevens, 2018 is not correct (L379), or (Zheng et al. Lamer et al. 2019; ...) in L44. Please improve throughout the manuscript.

We have reformatted the references at the end and revised and uniformed all citations in the text.

- L276: I understand that strong precipitation rates can introduce strong peaks, but this is the intermittent nature of rain, and I guess not a measurement error. Can you understand from your data how such strong precipitation rates are locally balanced? It would be very interesting to analyze this.

We agree with both reviewers on this. It is very difficult to estimate whether a very large spike in measured precipitation is due to splashing. For this reason, we have repeated the computations keeping all precipitation readings. The new Fig. 7 (now Fig. 5) shows the updated values, it can be seen that the contribution of precipitation to the moisture sink has increased especially in winter. As for the last question, I am not sure. I think looking at extreme precipitation may require the development of a more specific higher resolution dataset and probably an analysis of uncertainties of precipitation that uses all precipitation measurements at the site (perhaps adding the scanning precipitation radar?).

- L308: please specify what kind of filter is applied here.

Added "eliminating points beyond 2 standard deviations from the monthly mean" at Line 226.

L343: "The seasonality of the large-scale advection term is also the factor that determines the seasonality of the overall budget." --> Please clarify how you get to this statement. From Fig. 7 it seems that the seasonality in the LHF or precipitation terms is also very large.

The reviewer is correct. We corrected the statement: "The large-scale advection term is the largest moisture sink". At Line 332.

- L345ff: Does the magnitude of the monthly residuals depend on how much data was used per month? I.e. if only a few days of data could be used, it might not be surprising that the monthly budgets don't balance well. Please clarify.

It is likely that a smaller dataset would be more sensitive to the large noise. In our case with more than 5 years of data, after selecting marine cases and cleaning the dataset, there were between 700 and 1900 cases in each month. It is likely that a smaller dataset may require more attention to avoid biased results.

- L389: How do you interpolate 10-min profiles over 1 minute? Do you downsample the data?

It was just a linear interpolation between successive measurements. We are working on an improved high-resolution dataset of Raman lidar moisture profiles that will enable us to do a higher resolution analysis.

- Figures:
  - 11: Please specify what normalized height refers to in the right panel

    The height is normalized to the cloud base height. We now specify in the caption of Fig. 11 and 12.

  - 12: the figures are far too small and can hardly be read.

    We increased the font size of axis labels and legends.

- L511: I do not understand this sentence, please clarify: "The lower mid-tropospheric humidity during the winter months, together with turbulence (Ghate et al. 2021) point towards turbulence being the primary controlling factor rather than water vapor in determining cloudiness in the region."

We agree with the reviewer that the sentence was confused It was replaced with: "During winter months, turbulence, rather than water vapor, appears to be the primary controlling factor of cloudiness in the region (Ghate et al., 2021)." Line 500.

- L532: "Moist and dry patches present differences in vertical velocity with dry regions displaying more frequent downdrafts than moist regions immediately below the cloud base." ï   from Fig. 12 I'd say the opposite, please clarify.

Figure 12 a) shows that the frequency of occurrence of downdrafts in dry regions (blue dashed line) is about 50% between the normalized height of 0.6 and 1 compared to a frequency of downdrafts in the moist region (red dashed line) of about 35% in the same height range.

We changed the text to: "Moist and dry patches present differences in vertical velocity with dry regions (dashed blue lines in Fig. 12 a) displaying more frequent downdrafts than moist regions (red dashed lines) immediately below the cloud base."

**Typographical suggestions:**

L45: that that the water ...Done

L229 / L263: the total water mixing ratio is once written in normal and once in bold font – please harmonize. Done

L234: please use proper math formulation for the averaging brackets. Done

L279: maybe add after '... the entrainment rate **(see Sec. 4.2)**' Done

L303: ...balanced by local change in the boundary layer **height** (?) Done

L321: large-scale turbulence --> do you mean subsidence instead of turbulence? Yes, changed. And previously, largescale was written without '-', please make it consistent. We revised all instances

L322: that what reported --> than what was reported .. Done

L514: hear --> heat Done

---

## Author Response (AR2)

We have addressed the remaining reviewer's comments by adding in the supplemental material a comparison between the decoupling index used in this work and one proposed in Jones et al., 2011and showing specific cases in which the 2 indexes agree and disagree. We also addressed the minor revisions proposed.

I thank the authors for addressing my comments. However, in my view the revised version does not show convincing evidence that the issue of well-mixedness is resolved. A lot of questions remain open for me:

- Has this Decoupling Index been successfully used in other studies? If so, reference to previous work should be given.

Thank you for pointing this out, the following reference has now been added to the discussion (Line 204): Sena, E.T., McComiskey, A., Feingold, G.: A long-term study of aerosol–cloud interactions and their radiative effect at the Southern Great Plains using ground-based measurements, Atmos. Chem. Phys., 16, 11301–11318, doi:10.5194/acp-16-11301-2016, 2016.

- To help the readers' intuition, could the authors show example radiosonde profiles for the different DI thresholds? This could convince me that a DI of 0.25 is indeed a coupled boundary layer. We agree with the reviewer. Examples of profiles are now added to the supplemental material and mentioned in lines 207-211

- When I look at Fig. 2 of Albright et al. 2022, I find that the pdfs of LCL and the sub-cloud layer top (indicative of cloud base) are usually very close together also in the decoupled boundary layers of the downstream trades near Barbados, with DI's between about 0.03-0.15 (as estimated by eye). Furthermore, Albright et al. 2022 only apply the ML framework to the sub-cloud layer, and not the entire trade-wind boundary layer (i.e. excluding the cloud layer). How does this go together with your study? There are indeed decoupled cases in which the LCL and the cloud base are close, however these constitute a minority of the cases. When we compare the index used with the index in Jones et al., 2011 they disagree in about 12% of the cases. We have added more analysis in the supplemental material. In our case the ML approximation is necessary as there is no straightforward way to understand the controls of boundary layer water vapor apart from the mixed-layer budget framework. In the manuscript we have mentioned that this assumption is flawed and have discussed the ramifications of such approximation.

- How do the DI's depend on the averaging scale? LCL and CB might be closer together for individual radiosondes compared to averages over multiple sondes / longer times.

The classification of coupled and decoupled cases is not the main focus of the paper but is used here to evaluate the uncertainties due to the assumption of a mixed layer framework at the site. In this perspective a detailed analysis of the decoupling index is beyond the boundaries of this analysis, and we have not investigated these aspects.

- How is cloud base estimated from the Radiosondes? What formula / air parcel properties are used for the LCL computation?

The cloud based was determined from radiosondes relative humidity with a threshold of 0.95% and the LCL was computed from radiosondes using the approximation (eq.2 in Romps, 2017):

$$z_{LCL} = z + \frac{c_{pm}}{g} \left\{ T - 55K - \left[ \frac{1}{T - 55K} - \frac{\log(RH)}{2840K} \right]^{-1} \right\}$$

with z=100 m.

Romps, D. M.: Exact Expression for the Lifting Condensation Level, *Journal of the Atmospheric Sciences*, 74(12), 3891-3900, 2017.

- For evaluating the influence of the degree of decoupling on the budgets in L327-330, why use the subset with cloud base > 1.2 km, rather than a certain range of the DI?

Unfortunately, we can't calculate the decoupling index of the entire hourly dataset due to a lack of accurate thermodynamic profiles, that's why we calculated it exactly for the radiosondes and used the cloud base height as a proxy for the entire dataset.

**-Overall, I wonder whether other decoupling indicators could be more helpful to justify the assumption of coupled well-mixed boundary layers.**

Thank you for the suggestions, we repeated the calculations using the index developed in Jones et al., 2011 and added a comparison in the supplemental material.

**Other comments/typos:**

- L66: mention that Albright et al. 2022 focused on the downstream trades / near Barbados. -Done

- L73: LWP acronym not introduced - Done

- L114f: Say 'Quantities from ERA5' here and in the following, rather than 'from ECMWF'. -Done
- L116: of about -> by about-Done
- L148: better discussed -> discussed in more detail-Done
- L208: large-scale instead of mesoscale processes?- Done
- L210: q\_L -> q\_l Done
- L337: in within -> to within -Done

- L341: the moistening from the surface is primarily balanced by advection drying ( $\sim$ 50%) precipitation, and entrainment drying ( $\sim$ 25%) with the rest compensated through precipitation removal -> should precipitation be removed in the first place? -Yes, thank you-Done

- L380: one-minute -> 1-minute (or just consistently throughout the manuscript)-Done
- L387: is -> are-Done

- Table S2: I think the table would be more useful if some associated mean meteorological conditions would be shown along with it. E.g. T, q, wind, zLCL, zCB, zPBL, cloud cover, precip, LWP... Done

- Fig. 6: 'duration' is actually just wind speed. So why not write wind speed directly? And # of perturbations is just # of samples, as no minimum difference from the mean is specified for defining perturbations. I think it would be better to write explicitly what is shown, as otherwise readers might think it shows sth. different.

The y axis is the time that takes for the 10 km water vapor perturbation to pass over the site. The time depends on the wind speed but does not coincide with it. For example, if the wind speed is 5 m/s a 10 km perturbation lasts about 33 minutes. The perturbation is defined as a deviation from the background state defined with a 100 km lowpass filter. We added the correspondence to wind speed in the figure caption. Duration, rather than wind speed, may be more intuitive on a time/space reference.

- L414: why not sticking with the 10-km scale, why using 10-min now?

The perturbations are still defined using the same 10 km window and are shown in Fig. 6. From these we excluded those perturbations that lasted less than 10 minutes. From Fig. 6 these correspond to PWV deviations of less than 0.01 cm from the background state.

- Sec 5.2: would be valuable to mention / discuss the George et al. 2022 study here (https://doi.org/10.1002/essoar.10512427.1)

Thank you we had missed this relevant reference

- L461: 103 and 104 what? Updrafts per case/day? Then write e.g. '...in a total of 103 and 104 updrafts per case/day' Changed to: "total number of updrafts"